# Muscle niche-driven Insulin-Notch-Myc cascade reactivates dormant Adult Muscle Precursors in *Drosophila*

**Rajaguru Aradhya†‡, Monika Zmojdzian†, Jean Philippe Da Ponte, Krzysztof Jagla\***

Génétique Reproduction et Développement, INSERM U1103, CNRS UMR6293, Clermont-Ferrand, France

**Abstract** How stem cells specified during development keep their non-differentiated quiescent state, and how they are reactivated, remain poorly understood. Here, we applied a *Drosophila* model to follow in vivo behavior of adult muscle precursors (AMPs), the transient fruit fly muscle stem cells. We report that emerging AMPs send out thin filopodia that make contact with neighboring muscles. AMPs keep their filopodia-based association with muscles throughout their dormant state but also when they start to proliferate, suggesting that muscles could play a role in AMP reactivation. Indeed, our genetic analyses indicate that muscles send inductive dIlp6 signals that switch the Insulin pathway ON in closely associated AMPs. This leads to the activation of Notch, which regulates AMP proliferation via dMyc. Altogether, we report that *Drosophila* AMPs display homing behavior to muscle niche and that the niche-driven Insulin-Notch-dMyc cascade plays a key role in setting the activated state of AMPs.

**\*For correspondence:** christophe.jagla@udamail.fr

†These authors contributed equally to this work

**Present address:** ‡Sloan-Kettering Institute, Rockefeller Research Labs, New York, United States

**Competing interests:** The authors declare that no competing interests exist.

## Introduction

Stem cells exhibit a remarkable capacity to keep a quiescent undifferentiated state, and then, once activated, contribute to developmental growth or damage tissue regeneration. Over the years, lineage tracing and serial transplantation assays have confirmed the presence of stem cell populations in many tissues in both invertebrate and vertebrate organisms (for reviews, see [*Voog and Jones, 2010*; *Simons and Clevers, 2011*; *Jiang and Edgar, 2012*]). These populations include multipotent cells, such as hematopoietic stem cells (HSCs) which can give rise to a broad range of cell types (*Marquez-Curtis et al., 2011*), and committed precursors, such as the satellite cells required for postnatal growth and the repair of a specific tissue, i.e. skeletal muscle (*Buckingham and Montarras, 2008*). Importantly, stem cell activity and capacity to maintain tissue homeostasis depend on a specialized microenvironment called the niche. The stem cell niche was first documented in *Drosophila* (*Xie and Spradling, 2000*) but it is now widely accepted that all adult stem cells reside within a niche that retains them and regulates their behavior (*Voog and Jones, 2010*). Niches range in size and complexity (*Morrison and Spradling, 2008*). They may house a single stem cell, like the follicle stem cell (FSC) niche (*Nystul and Spradling, 2007*), or more than 10 germ stem cells (GSCs), like the testis niche (*Wallenfang et al., 2006*). Niches may also occupy a single spatially invariant location throughout adult life (e.g. the GSC niche in *Drosophila*), or instead form a complex system of multiple niches distributed throughout tissues (e.g. HSC niches) (for a review, see *Morrison and Spradling, 2008*). Like HSCs, which are able to reside in alternative niches, muscle stem cells (satellite cells) are scattered under the basal laminae of myofibers, which host them and contribute to their niche (*Collins et al., 2005*; *Kuang et al., 2007*). However, the cellular and molecular mechanisms that control niche occupancy by stem cells remain poorly understood. Recent work on

**eLife digest** Muscles experience wear and tear over our lifetimes and therefore need to be regularly repaired and replenished by new cells. These cells are produced by stem cells, which often reside in a special microenvironment called the stem cell niche. This niche may also contain support cells that produce signals to attract stem cells and then maintain them in a dormant state. When the muscle is damaged, its resident stem cells are activated so that they divide to produce new cells. Understanding how this happens is an important goal for regenerative medicine, but many of the details remain unclear.

In fruit flies, stem cells called adult muscle precursor cells (or AMPs for short) lie dormant in the embryo and larva, but are then activated to form the muscles of the adult fly. These cells share many features with the muscle stem cells of mammals, which prompted Aradhya, Zmojdzian et al. to use them as a model to investigate how stem cells find their niche and are later activated.

For the experiments, the AMPs in fruit fly larvae were labelled with a fluorescent protein. Aradhya, Zmojdzian et al. observed that these cells produce long extensions that connect them to each other, to nearby muscle and to nerve cells. During development, these extensions are gradually lost until they contact only the muscles that are closest to the AMPs, which indicates that these muscles provide a niche for the AMPs and are perhaps involved in their activation.

Further experiments show that neighbouring muscles do indeed help to activate AMPs, as they produce a signal that activates a cell communication system called the insulin pathway inside the AMPs. Insulin signalling – which is sensitive to the availability of nutrients in the body – turns on another signalling pathway, called Notch, that then stimulate the AMPs to divide. Aradhya, Zmojdzian et al. propose that this signalling cascade might help to ensure that AMPs are only activated at the right time in development. The next step is to find out whether stem cells in human muscles are activated in a similar way.

myogenic progenitor cells, which ensure developmental muscle growth but also provide a source of cells that adopt satellite cell position, has shed new light on this issue (*Bröhl et al., 2012*). It has been shown that Notch signaling is required for the homing of emerging satellite cells by stimulating them to produce basal lamina, thus promoting their adhesion to myofibers (*Bröhl et al., 2012*). Crucially, if the homing process is impaired, satellite cells are unable to receive inductive signals and thus efficiently contribute to muscle growth and regeneration. This makes gaining further insight into the homing of stem cells and their responsiveness to the signals emanating from the niche a key challenge.

We have previously characterized the *Drosophila* muscle stem cells called adult muscle precursors (AMPs) that emerge during mid-embryogenesis and express muscle progenitor-specific markers such as the b-HLH transcription factor Twist (*Figeac et al., 2007*, *2010*). The AMPs lie dormant during embryonic and most of larval life but once activated they will proliferate to provide a source of myoblasts that ensure adult muscle growth and the regeneration of a subset of thoracic flight muscles. We also followed AMP cells in vivo using membrane-targeted GFP, and found that AMPs send out long cellular processes, and are interconnected (*Figeac et al., 2010*). Interestingly, the capacity to send out cytoplasmic extensions and make interconnections has also been documented for quiescent satellite cells sited on myofibers (*Tavi et al., 2010*). All these features make AMPs similar to vertebrate satellite cells, prompting us to analyze their homing behavior and the mechanisms that drive their activation and exit from the dormant state.

Our data show that emerging AMPs, in addition to long cellular projections, also send out thin filopodia that link them to the neighboring muscles, which behave as AMPs cell niche. We provide genetic evidence that muscles act via dIlp6 to switch the insulin pathway ON in AMPs and initiate AMP reactivation. This leads to a Deltex-involving activation of Notch, which positively regulates AMP proliferation via dMyc.

## Results

### AMPs display homing behavior and become tightly associated with neighboring muscles

AMPs are specified at embryonic stage 12 and then remain quiescent and undifferentiated until the mid-second larval instar (*Bate et al., 1991*). We showed in earlier work that soon after their specification, embryonic AMPs form an interconnected network via long cytoplasmic extensions (*Figeac et al., 2010*). A similar feature has also been reported for the quiescent vertebrate satellite cells, which are connected to each other and to the adjacent muscle through thin cytoplasmic extensions termed 'tunneling nanotubes' (*Tavi et al., 2010*). To examine the dynamics of AMP cell morphology and behavior in more detail, we generated an AMP sensor line, m6-gapGFP (see Materials and methods) that enabled us to visualize the shapes of AMPs in vivo. We focused our analyses on the abdominal AMPs, which when quiescent form a repeat pattern of six cells per hemisegment (*Figeac et al., 2010*). Initially, at embryonic stage 12, AMPs appear spherical in shape and are separated from each other (*Figure 1—figure supplement 1A*), but a closer view (*Figure 1A*) shows that they send out numerous thin filopodia around their surface. This 'sensing behavior' also persists in later embryonic stages (*Figure 1B,C*), in which AMPs become more elongated and send out long cytoplasmic extensions (*Figure 1C* and *Figure 1—figure supplement 1B*) to form an interconnected network (*Figeac et al., 2010*). The long cellular processes follow the main neural branches of the peripheral nervous system (PNS) (*Figure 1C'*, arrows), while the short filopodia display dynamic and irregular patterns and seem not to be attracted by the PNS nerves (*Figure 1C'*, arrowheads).

As the embryonic AMPs are the immediate neighbors of somatic muscles (*Figeac et al., 2007*, *Figeac et al., 2010*), we co-visualized the AMPs and the adjacent muscle cells by two-color live cell imaging (*Figure 1D–G* and *Video 1*). Our data reveal that the small filopodia sent out by AMPs made contact with surrounding muscles (*Figure 1D-G'* and *Video 1*) and by embryonic stage 16 had become tightly associated with neighboring muscles.

To characterize AMP localization with respect to internal versus external muscles, we created Z-stack movies of m6-gapGFP embryos (*Video 2* and *Video 3*). We observed that AMPs associate with several rather than just one particular muscle layer. For example, the lateral AMPs extend from the external to internal layer. The posterior lateral AMP (arrows in *Video 2* and *Video 3*) lies externally over the SBM, and is seen at the same level as the external lateral LT muscles. The anterior lateral AMP (arrows in *Video 2* and *Video 3*) lies more internally, mainly at the level of internal lateral muscles LO1 and SBM.

To test whether AMP interactions with specific muscles are underpinned by filopodia dynamics, we performed a time-lapse experiment at embryonic stage 15 (*Video 4*). The developmental time-window chosen corresponds to the homing period in which AMPs actively send filopodia and attempt to make contact with target muscles. To follow the number of filopodia and the direction of their projections, we labeled the extremities of all filopodia at each time-point (indicated by yellow circles in *Video 4*). We focused on lateral AMPs and found that they send out filopodia non-randomly and mainly in anterior-dorsal directions, which correlate with the location of SBM and LO/LT muscles to which lateral AMPs are connected. Thus, the filopodia are projected mainly in the direction of targeted muscles, ultimately enabling a subset of them to stabilize (arrowheads in *Video 4*).

However, reaper-induced muscle ablation experiments (*Figure 2A–D*) revealed that there is some plasticity in AMP-muscle interactions. For example, in segments with ablated dorsal and dorso-lateral muscles (*Figure 2B*), some of the dorso-lateral AMPs interacted with remaining LT1/2 muscles (arrowhead in *Figure 2B*) left unconnected in the wild-type context (arrowhead in *Figure 2A*), whereas dorsal AMPs were unable to do so and adopted rounded shapes (yellow arrowheads in *Figure 2B*). This finding indicates that dorsal-ventral positional information restricts AMP–muscle contacts. Moreover, we observed AMP cell loss (asterisks in *Figure 2*) correlating with the severity of muscle ablation phenotypes (compare *Figure 2B,C and D*), which suggests that AMP interactions with muscles are important for their survival.

In addition to the filopodia-based contacts, AMP cell bodies appear directly associated with particular muscle fibers. For example, posterior lateral AMP extends over the SBM muscle (*Figure 1F, F'*). This cell-body contact involves the AMP and muscle originating from the common muscle progenitor (*Jagla, et al., 1998*), suggesting that a shared lineage might facilitate interactions. Both the filopodia and cell body involving AMP-muscle connections are decorated by punctate expression of

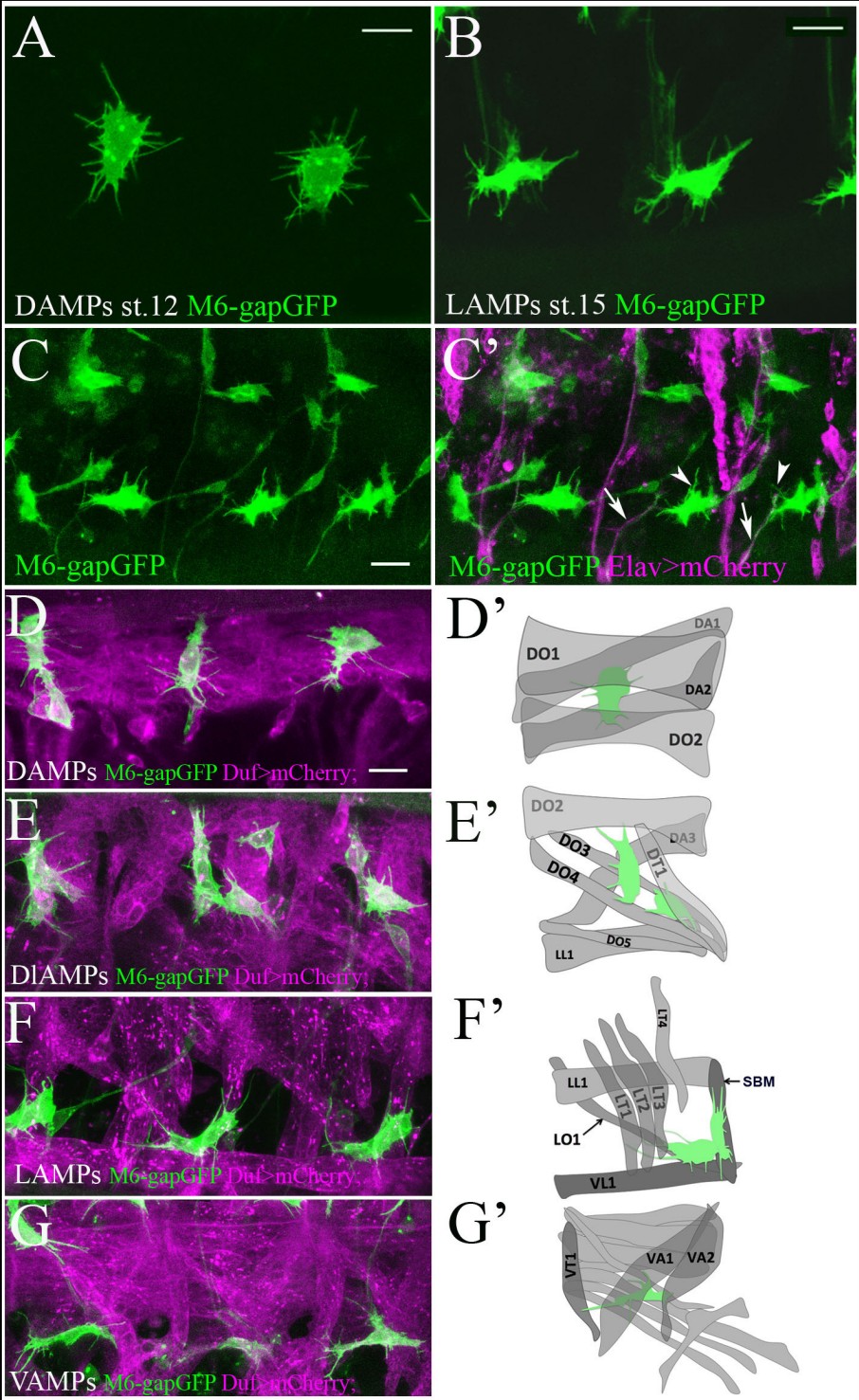

**Figure 1.** Quiescent AMP cells are tightly associated with surrounding muscles. (**A, B**) A zoomed view of quiescent dorsal (**A**) and lateral (**B**) AMPs bearing numerous thin filopodia. (**A**) Newly-specified AMPs at embryonic stage 12 display a random pattern of filopodia. (**B**) Mid-stage embryo AMPs become elongated and send out filopodia in an directionally-oriented way. Filopodia pattern of AMPs in m6-gapGFP embryos was revealed by anti-GFP staining of membrane-targeted GFP. (**C, C'**) A lateral view of three hemisegments of stage-15 embryo from the sensor driver line m6-gapGFP; Elav-GAL4; UAS-mCD8mCherry, driving mCherry with a membrane localization signal in all neurons. Arrows point to cytoplasmic extensions connecting the AMPs (green) and aligned with the PNS nerves (magenta). Arrowheads denote thin filopodia that are not connected to the PNS nerves. (**D–G**) Dual-color in vivo views of three hemisegments of stage-15 embryos from the m6-gapGFP; Duf-GAL4; UAS-mCD8mCherry line. mCherry (magenta) reveals embryonic muscles and GFP (green) reveals AMPs. Dorsal (**D**), dorsolateral (**E**), lateral (**F**) and ventral (**G**) groups of AMPs are shown. Note that AMPs

*Figure 1 continued on next page*

*Figure 1 continued*

connect to the embryonic muscles with numerous filopodia. (D′–G′) Schemes represent all observed AMP-muscle connections. AMPs connect to a defined set of muscles. (D′) Dorsal AMP connects to DO1 and DA2 and optionally to DA1 and DO2. (E′) Dorsolateral AMPs connect to DT1, DO3, DO4 and DO2. (F′) Lateral AMPs connect to SBM, LT1, LT2, LT3 and to LO1 and VL1. (G′) Ventral AMP interacts with VA2, VT1 and VA1. Scale bar in (A, B): 4 microns, in (C–G): 9 microns.

The following figure supplement is available for figure 1:

**Figure supplement 1.** Segmental pattern of embryonic AMPs.

a-PS1 and βPS integrin (*Figure 2E–H′*, *Videos 5* and *6*). The first α-PS1 punctate signals associated with the AMP cell body appear at late-stage 14 (arrowhead, *Figure 2E,E′*) and are progressively enriched at stages 15 and 16 (*Figure 2F, G′*, *Videos 5* and *6*). Punctate α-PS1 patterns were associated mainly with the AMP cell bodies (arrowheads in *Figure 2F,G′*, *Videos 5* and *6*) but were also seen to be aligned with filopodia (arrows in *Figure 2F,G′*, *Videos 5* and *6*).

Similarly distributed but more discrete β-PS dots were also detected from stage 15 (*Figure 2H,H′*). As filopodia are highly dynamic structures, we posit that integrins mark filopodia subsets that are making contact with target muscles and in the process of stabilization. This hypothesis is supported by in vivo analysis of filopodia dynamics showing that some filopodia indeed get stabilized (arrowheads in *Video 4*).

Taken together, these observations suggest that AMPs like emerging satellite cells (*Bröhl et al., 2012*) display homing behavior to muscle niche.

## AMPs keep contact with muscle niche during their reactivation

During larval stages, body wall muscles grow rapidly and increase several times in size, which

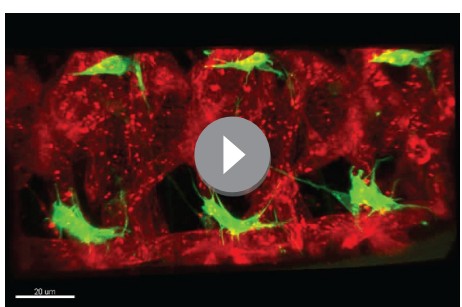

**Video 1.** A 3D-reconstruction of the lateral AMPs and surrounding muscles of the stage 15 embryos M6-gapGFP; Duf-GAL4; UAS-mCD8mCherry embryos. Note that all the small filopodia sent by AMPs (green) connect to the muscles (red).

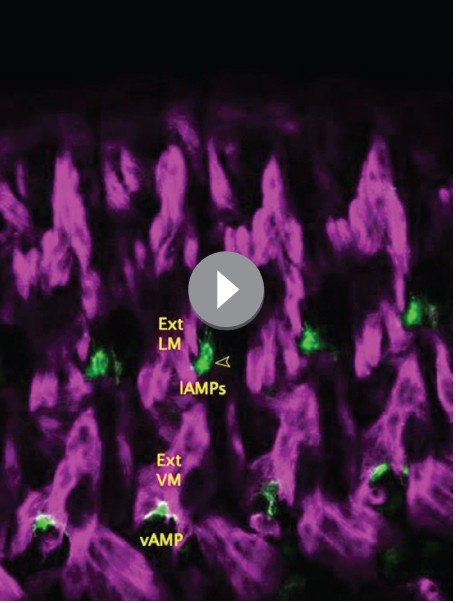

**Video 2.** AMPs localization with respect to external/internal muscle layers in stage 14 embryo. A Z-stack movie of M6-gapGFP embryos stained for muscles (β3-tubulin - magenta) and AMPs (GFP - green). The ventral, lateral, dorso-lateral and dorsal AMPs have distinct internal/external locations. The lateral AMPs (lAMPs) extend from the external to internal layer. The posterior lAMP (arrowhead) lies the most externally and is seen at the same optical level that the external lateral muscles (ExtLM: LT muscles). The anterior lAMP (arrow) lies more internally, mainly at the level of internal lateral muscles (IntLM: LO1 and SBM). The ventral AMPs (vAMPs) are located in between external ventral muscles (ExtVM: VA1 and VA2) and intermediary ventral muscles (ImVM: VO3-VO6) but they send cellular extensions externally and are seen at the level of VA1 and VA2. The dorso-lateral AMPs (dlAMPs) are clearly located under the external DT1 and lie mainly in between the intermediary dorsal muscles (InDM: DO3 and DO4) and internal dorsal muscles (IntDM: DA3). Finally, the dorsal AMPs (dAMPs) are located in between the external (ExtDM: DO1 and DO2) and internal dorsal muscles (IntDM: DA1, DA2). Note: view movie frame by frame to appreciate AMPs positioning and to see corresponding annotations.

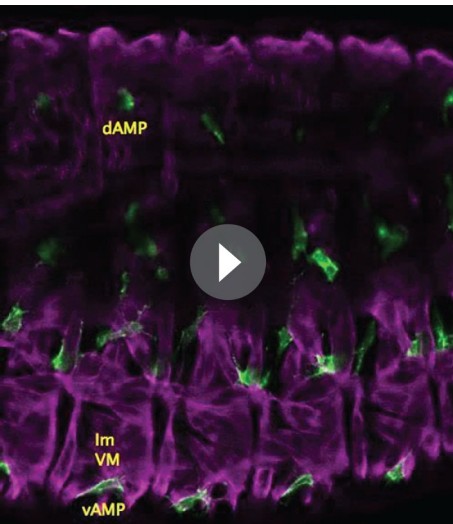

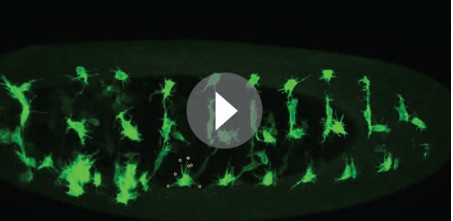

**Video 3.** AMPs localization with respect to external/internal muscle layers in stage 15 embryo. A Z-stack movie of M6-gapGFP embryostained for muscles (β3-tubulin - magenta) and AMPs (GFP - green). Refer to the legend of *Video 2*. Note: view movie frame by frame to appreciate AMPs positioning and to see corresponding annotations.

**Video 4.** Lifetime imaging of AMPs filopodia dynamics. A lateral view of M6-gapGFP stage 15 embryo is shown. Z-stacks were taken each 1 min during the period of 35 min. The filopodia of lateral AMPs from one segment were annotated. To follow the number of filopodia and the direction of their projection we labeled the extremities of all filopodia at each time point (indicated by yellow open circles). We found that the lateral AMPs send filopodia non-randomly in directions, which correlate with location of muscles to which they are connected by stage 16. We observed that 6 to 9 filopodia are visible at each time point. Some of filopodia appear more stable than others (indicated by arrowheads). Filopodia projecting in dorsal direction (denoted by the arrowhead) does not acquire stable state. Note: A frame by frame viewing of the movie will allow to count number of filopodia per time point and appreciate filopodia extension and retraction events.

raises the question of whether AMPs keep associated with the growing larval muscles and whether the long cellular extensions interconnecting the AMPs persist. We found that the interconnecting cellular processes are present in first-instar larval AMPs (*Figure 3—figure supplement 1A*) but are then progressively lost, becoming undetectable in second instar larvae (*Figure 3—figure supplement 1B*). However, the filopodia- and cell body-based contacts of AMPs with neighboring muscles persist along larval life until AMP reactivation (*Figure 3B,C*). Compared with the late embryonic stage (*Figure 3A*), quiescent AMPs at early second larval instar (*Figure 2B*) project relatively few filopodia. As shown for lateral AMPs (*Figure 3B*), they adopt highly elongated shapes with two long cellular protrusions that extend and follow growing muscle. At this stage, the lateral AMPs restrict their contact to the two closest muscle neighbors, i.e. SBM and LO1, illustrating preferential interactions with these two muscles already seen in embryos.

The observation that the AMPs maintain elongated shapes with long cellular extensions aligning neighboring muscles until the beginning of their reactivation (*Figure 3C*) suggests that the association of AMPs with muscle niche could play a role in their exit from the quiescent state. After a few rounds of proliferation, AMPs adopt more rounded shapes (*Figure 3D*) but stay closely associated to each other and to the muscle niche (*Figure 3D*).

## Insulin and TOR signaling pathways positively regulate the reactivation of AMPs from their dormant state

The AMPs that are at the origin of adult *Drosophila* muscles are quiescent from mid-embryogenesis until the mid of the second larval instar (*Figure 3*). The progenitors of the fly brain, the neuroblasts, also behave quiescently during development, and it has been reported that their exit from the dormant state is subject to a nutritional checkpoint involving the TOR pathway and that glial cell-derived Insulin signals are required to initiate their proliferation (*Chell and Brand, 2010*, *Sousa-Nunes et al., 2011*). We observed that AMPs stay quiescently in third-instar larvae growing in nutrient-restricted conditions (*Figure 3—figure supplement 1C*), suggesting that similar mechanisms could also drive

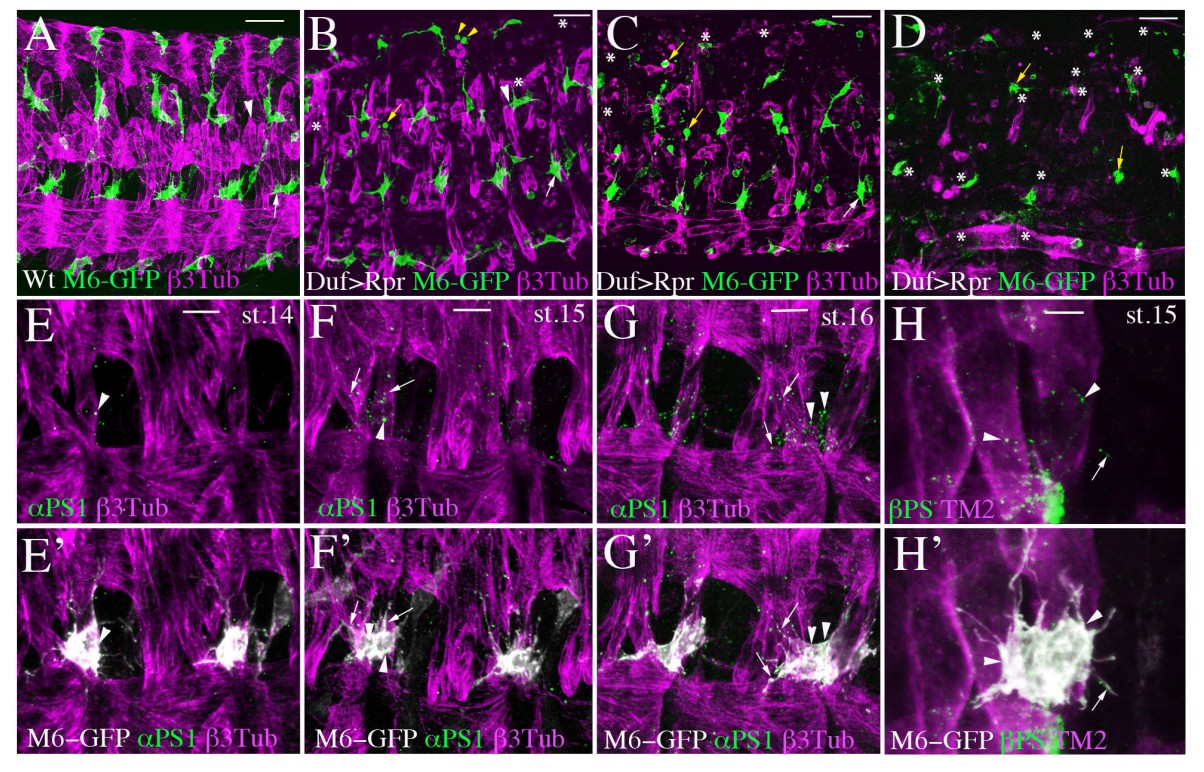

**Figure 2.** AMP-muscle connections display spatially-restricted plasticity and are decorated by integrin expression. (**A**) A wild-type view of AMPs and muscles from mid-stage m6-gapGFP embryo. (**B–D**) Similar views from m6-gapGFP;Duf-GAL4;UAS-Rpr embryos with (**B**) weak, (**C**) intermediate, and (**D**) strong muscle ablation phenotypes. In segments with partial loss of lateral muscles, the anterior lateral AMP, which normally extends anteriorly (white arrow in A) remained tightly associated with the posterior lateral AMP and interacted mainly with SBM muscle – (white arrows in B and C). In segments with loss of dorsal and dorso-lateral muscles and with some lateral muscles persisting, (**B**) the dorso-lateral AMPs interacted with remaining lateral muscles (arrowhead in B) to which they do not connect in the wild-type context (arrowhead in A). This indicates a degree of plasticity in AMP connections. In segments with a pronounced loss of dorsolateral and lateral muscles (B and C), the dorsal and dorso-lateral AMPs adopted rounded shapes (yellow arrows) and were unable to migrate to other segments or to the ventral region where muscles were still present. In embryos with total muscle ablation, the majority of remaining AMPs adopted rounded shapes (yellow arrows in **D**). The number of AMPs detected was drastically reduced (asterisks indicate lacking AMPs). (**E–H'**) Zoomed views of lateral AMPs stained for (**E-G'**) α-PS1 and (**H, H'**) βPS integrin. The first α-PS1 dotty signals associated with AMPs appear at late-stage 14 (**E, E'**) and are progressively enriched at stages 15 and 16 (**F-G'**). A punctate α-PS1 pattern is seen, associated with AMP cell bodies (arrowheads) but also aligned with filopodia (arrows in **F–G'**). A similar β-PS1 pattern denoted by arrows and arrowheads is also observed, starting from embryonic stage 15 (**H-H'**). Scale bars in (**A-D**): 30 microns; in (**E-G**): 10 microns; in (**H**): 6 microns.

their reactivation. To determine the influence of different signaling components on AMP proliferation, we used the AMP-specific driver M6-Gal4 to analyze the impact of signal deregulation by counting the AMPs in synchronized larvae at mid-third instar (*Figure 4*, *Figure 4—figure supplement 1* and *Figure 4—source data 1*).

The AMP-targeted expression of PTEN, which negatively regulates Insulin signaling or TOR inhibitors (TSC1, TSC2), results in dramatically lower numbers of AMPs (*Figure 4B,E,J*, *Figure 4—figure supplement 1* and *Figure 4—source data 1*). Conversely, we found higher numbers of AMPs following the overexpression of InRCA, a constitutively activated form of the Insulin receptor, or Rheb, the positive modulator of the TOR pathway (*Figure 4D,J*, *Figure 4—figure supplement 1* and *Figure 4—source data 1*). However, these same components were not sufficient to drive AMP exit from quiescence in the embryonic stages (*Figure 4—figure supplement 1* and *Figure 4—source data 1*) and only managed to accelerate AMP entry into proliferation from the mid to early second larval instar. This suggests that Insulin/TOR pathways are part of a more complex regulatory cascade driving the reactivation of dormant AMPs in a developmental time-window restricted to mid-larval stages.

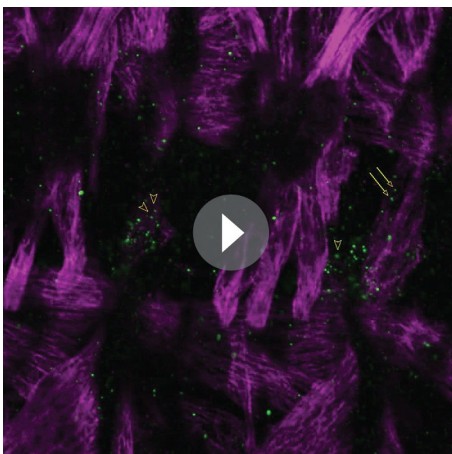

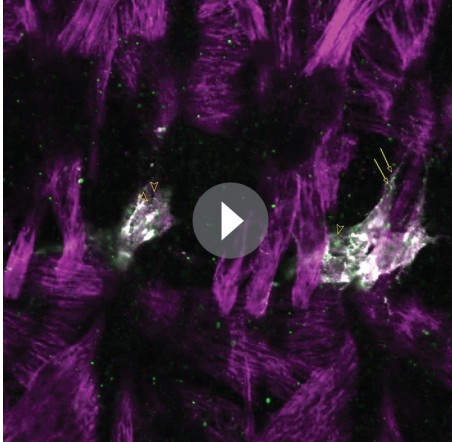

**Video 5.** αPS1 integrin decorates AMP cell bodies and filopodia projections. Two-channel Z-stack movie showing punctate decoration of lateral AMPs by αPS1 integrin at embryonic stage 16. The AMP cell bodies (arrowheads) and filopodia-associated αPS1 dots (arrows) are annotated.

**Video 6.** αPS1 integrin decorates AMP cell bodies and filopodia projections. Three-channel Z-stack movie showing punctate decoration of lateral AMPs by αPS1 integrin at embryonic stage 16. The AMP cell bodies (arrowheads) and filopodia-associated αPS1 dots (arrows) are annotated. Note: view *Videos 5* and *6* movies in parallel and frame by frame to follow αPS1 dots associated with the AMPs and corresponding annotations.

## A novel role for Notch and dMyc in promoting AMP proliferation

We previously reported that a regulatory element of *Enhancer of split m6 (E(spl)m6; m6)* gene carrying potential Supressor of Hairless (Su (H)) binding sites (*Rebeiz et al., 2002*) drives expression specifically in quiescent AMPs (*Figeac et al., 2010*). This suggests that the Notch pathway is activated in embryonic AMPs, but the question of whether it plays a role in setting quiescent versus proliferative AMP state remained unsolved. Here, we showed that m6-Gal4-targeted expression of the Notch intracellular domain (NICD) in embryonic AMPs does not alter their quiescence in embryos (*Figure 4—figure supplement 1*). However, in the same NICD context, there were significantly higher numbers of AMPs in third-instar larvae, suggesting that Notch reactivates AMPs in larval stages and promotes their proliferation (*Figure 4F,J*, *Figure 4—figure supplement 1* and *Figure 4—source data 1*). The reduced number of AMPs in larvae with m6-driven Notch attenuation (*Figure 4G,J*, *Figure 4—figure supplement 1* and *Figure 4—source data 1*) further supports this observation. As the decrease in AMP numbers in a Notch-RNAi context has not been associated with a reduced level of GFP driven by the same m6 regulatory element (*Figure 4—figure supplement 2*), we hypothesize that a low Notch level is sufficient to maintain m6 activity but we cannot rule out a possibility that perdurance of Gal4 and GFP in larval stages plays role as well.

In vertebrate satellite cells, Notch regulates the asymmetric divisions controlling the number of reserve muscle stem cells that, at the end of the cell cycle, sit on muscles and remain undifferentiated (*Kuang et al., 2007*). The activated AMPs do not seem to divide asymmetrically, as they did not increase in number in the *numb* RNAi context (*Figure 4—figure supplement 1*) and no reactivated AMPs were found to express the asymmetric cell division marker Numb-CD2-GFP (*Rebeiz et al., 2011*) (*Figure 4—figure supplement 1*). Notch thus plays a novel role in AMPs, promoting their proliferation without driving asymmetric cell divisions.

Besides the Insulin and Notch pathways, a recent study (*Li et al., 2012*) found that in vertebrates the transcription factor Myc is also involved in regulating myoblast proliferation during muscle development and regeneration. This prompted us to test whether Myc also regulated the proliferation of AMPs. Indeed, dMyc acts as a positive regulator of AMP reactivation (*Figure 4H–K*, *Figure 4—*

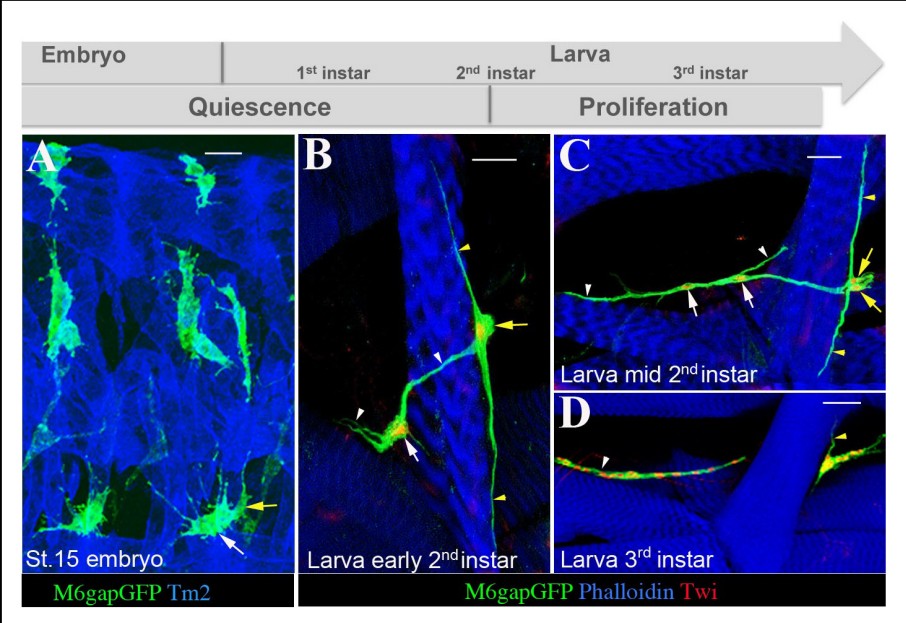

**Figure 3.** AMPs stay connected to surrounding muscles until reactivation. (**A**) A dorso-lateral view of two hemisegments in mid-stage embryo showing lateral, dorso-lateral and dorsal AMPs (green) and embryonic muscles (blue). Two lateral AMPs (white and yellow arrows) send numerous filopodia to lateral muscle fibers. Note that one of the lateral AMPs (yellow arrow) extends along the segment border muscle (SBM). (**B**) A zoomed view of two lateral AMPs from the early second larval instar. The AMP indicated by the yellow arrow stays connected to the SBM and sends two long cellular extensions (yellow arrowheads) along the SBM. The second lateral AMP (white arrow) still produces filopodia (white arrowheads) linking it with the SBM and the LO1 muscle. The number of filopodia-based AMP-to-muscle connections is reduced compared to embryonic stages. Nuclei of AMPs (red) are revealed by anti-Twi staining. (**C**) A similar view of lateral AMPs from mid-second larval instar undergoing first cellular division. Note that the reactivated AMPs indicated by two white and two yellow arrows keep their extended shapes and filopodia-based connections (white and yellow arrowheads) to the SBM and LO1 muscles. (**D**) Proliferating lateral AMPs from third instar larva labeled with anti-Twist (red) to reveal their nuclei and anti-GFP (green) to reveal their shapes. The remaining cellular extension (yellow arrowhead) is still shown connecting one of lateral AMPs to the SBM muscle. The cells originating from the AMP connected to the LO1 muscle are aligned along this muscle (white arrowhead). Note that proliferating AMPs form clusters of tightly-associated cells. Scale bars in (**A**): 12 microns; in (**B, C**): 25 microns; in (**D**): 36 microns.

The following figure supplement is available for figure 3:

**Figure supplement 1.** Larval AMPs adapt their shapes and keep associated to rapidly growing muscles.

*figure supplement 1* and *Figure 4—source data 1*), thus providing evidence that the proliferation of muscle stem cells is regulated by the same set of genes in both *Drosophila* and vertebrates.

## Notch acts downstream of the Insulin pathway and regulates the proliferation of AMPs via dMyc

To gain a better understanding of the functional link between Insulin, Notch and dMyc in AMP cell behavior, we analyzed their activity in reactivated AMPs. We first tested whether activation of the Insulin pathway was correlated with AMP proliferation. It has been reported that the plekstrin homology (PH) domain containing t-PGH protein binds specifically to phosphatidylinositol-3,4,5-P3 (PIP3) and, if localized to plasma membrane, indicates PI3K/Insulin pathway activity (*Britton et al., 2002*). We thus used transgenic t-PGH larvae to follow subcellular t-PGH localization in reactivated AMPs. The data show that t-PGH is specifically recruited to plasma membrane in AMPs that undergo cell divisions (*Figure 5A- A''* and *Figure 5—figure supplement 1*). Importantly, the proliferating AMPs also display high levels of intracellular Notch and nuclear dMyc (*Figure 5B–B''*), showing that activation of the Insulin pathway and increased Notch and dMyc levels correlate with the reactivated AMPs state.

We then tested whether the Insulin pathway acted upstream of Notch and dMyc. First, we quantified the level of Notch and dMyc in the individual lateral AMPs, and found that the mean fluorescent

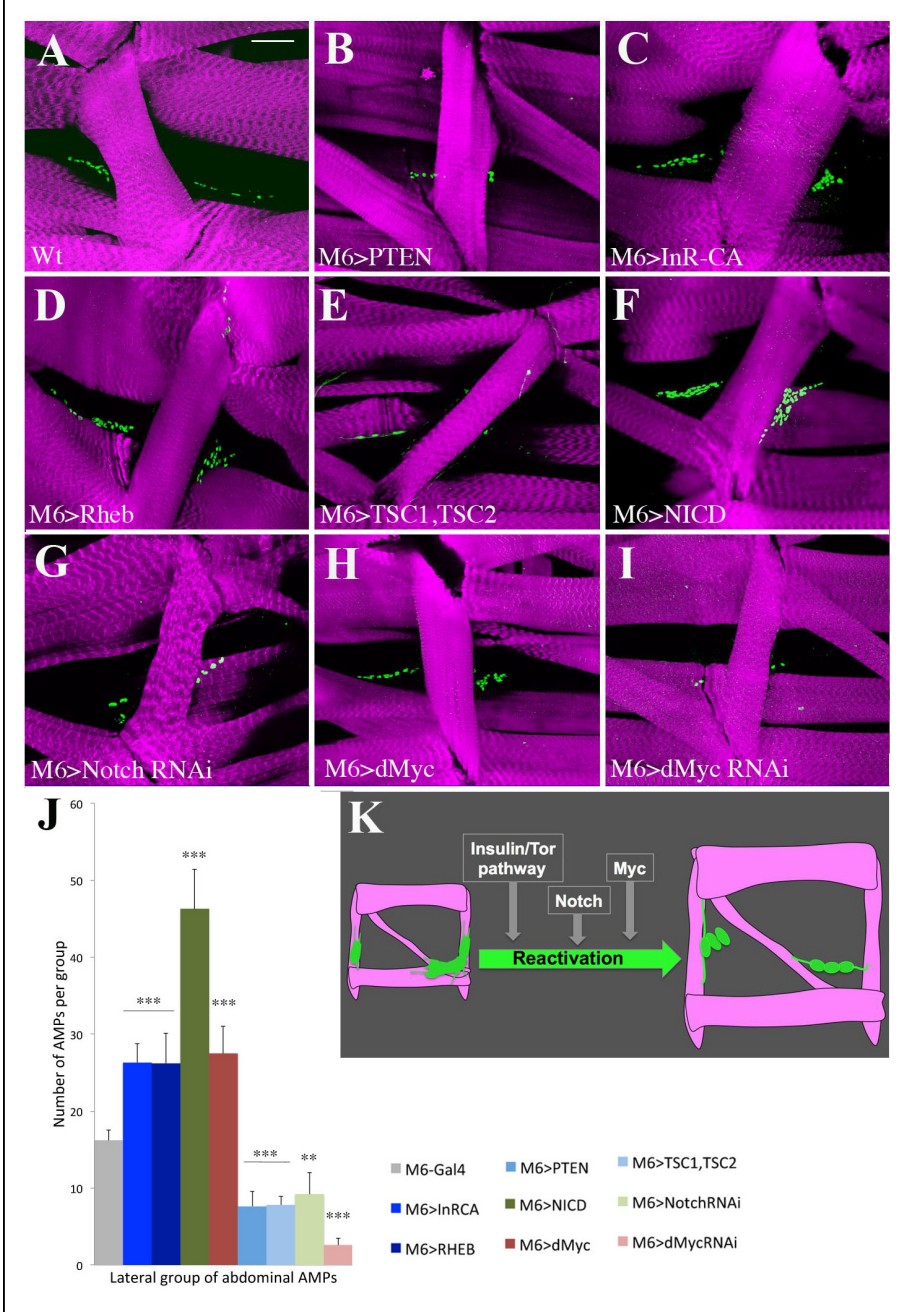

**Figure 4.** Insulin/TOR and Notch pathways control AMP reactivation in larval stages. (A–I) Flat preparations of the mid-stage matched third-instar larvae stained for Twist (green) labeling AMP nuclei and stained for Phalloidin (magenta) labeling the larval muscles. The abdominal lateral group of AMPs is shown in (A) representative control larva (M6-Gal4) and (B–I) in larvae with modified Insulin, TOR, Notch and Myc expression. M6-Gal4 driver is used to AMP-specifically drive the expression of: (B) PTEN, an inhibitor of the Insulin pathway; (C) InR-CAAX, a constitutively activated form of insulin receptor; (D) RHEB, an activator of the TOR pathway; (E) TSC1, TSC2, a complex of two proteins that inhibits the TOR pathway; (F) NICD, Notch intracellular domain that constitutively activates the Notch pathway; (G) dsRNA against Notch transcript; (H) overexpression of dMyc; (I) dsRNA against dMyc transcript. (J) Graphical representation of the mean number of lateral AMPs in the different genetic contexts shown in (A–I). (***) indicates $P \leq 0.001$. Scale bar: 36 microns. (K) A scheme illustrating the promoter influence of Insulin and Notch pathways and Myc on AMP reactivation.

The following source data and figure supplements are available for figure 4:

**Source data 1.** Table showing mean number of dorsal, lateral and ventral AMPs in the abdominal segments from the genotypes shown in *Figure 4A–I* and *Figure 4—figure supplement 1*.

*Figure 4 continued on next page*

*Figure 4 continued*

**Figure supplement 1.** Influence of Insulin, TOR, Notch and Numb on AMP cell number in larval stages and in embryos.
**Figure supplement 2.** M6-Gal4 driver keeps active in Notch attenuation context.

intensity representing Notch and dMyc protein levels was increased in the InRCA context (*Figure 5C,C'',J,K* and *Figure 5—source data 1*) compared with wild-type (*Figure 5B,B'',J,K* and *Figure 5—source data 1*) and was significantly lower in the AMPs expressing the negative regulator of Insulin signaling PTEN (*Figure 5J,K* and *Figure 5—source data 1*). The Insulin pathway thus positively regulates Notch and dMyc during AMP reactivation. Next, using the same approach, we tested dMyc protein levels in AMP-targeted gain and loss of Notch function, and found that Notch increased nuclear dMyc levels (*Figure 5D,D''*, *Figure 5—figure supplement 1* and *Figure 5—source data 1*), suggesting that Notch acts upstream of dMyc. The sum of these observations prompted us to determine whether dMyc acted as an effector of Insulin and Notch pathways, and whether Notch functioned downstream of Insulin during AMP proliferation. We found that lowering Notch or dMyc levels in AMPs expressing InRCA led to an attenuation of the AMP overproliferation phenotype, indicating that both Notch and dMyc act downstream of the Insulin pathway (*Figure 5E–G,L, M*, *Figure 5—figure supplement 1* and *Figure 5—source data 1*). We also observed that the increased number of AMPs generated in the NICD-overexpressing context was dMyc-dependent (*Figure 5H,I,L, M*, *Figure 5—figure supplement 1* and *Figure 5—source data 1*), indicating that dMyc acts as an effector of Notch in reactivated AMPs. Taken together, these data establish an Insulin-Notch-dMyc cascade governing the exit of AMPs from the dormant state and promoting their proliferation (*Figure 5M*). However, as NICD overexpression induces higher numbers of AMPs than InRCA overexpression (*Figure 5L*, *Figure 5—figure supplement 1* and *Figure 5—source data 1*), we cannot rule out the possibility that Notch also acts in an InR-independent way.

## Deltex-involving activation of Notch downstream of the Insulin pathway promotes AMP proliferation

The increased levels of intracellular Notch in AMPs expressing the activated form of Insulin receptor (InRCA) suggested that Insulin promotes Notch pathway activity during AMP reactivation. A similar observation was recently reported in *Drosophila* intestinal stem cells in which proliferation and differentiation is finely tuned by the interplay between the Insulin and Notch pathways (*Foronda et al., 2014*). However, the issue of whether the Insulin-dependent regulation of Notch involves conventional Delta/Serrate signal transduction has never been addressed. We thus tested whether the attenuation of Delta or Serrate in the muscle or PNS cells with which AMPs are associated impacts on AMP proliferation (*Figure 6—figure supplement 1* and *Figure 6—source data 1*). We found that knocking down Notch ligands in direct AMP cell neighbors has no effect on AMP reactivation (*Figure 6—figure supplement 1* and *Figure 6—source data 1*). Similarly, expressing in AMPs a dominant-negative form of Notch receptor (ECN) devoid of intracellular domain and known to efficiently repress canonical Notch signaling (*Rebay et al., 1993*) had no effect on AMP cell numbers (*Figure 6—figure supplement 1* and *Figure 6—source data 1*), suggesting that Notch activation in AMPs could occur in a ligand-independent way. To further explore the Insulin-Notch pathway linkage, we tested the expression of the ubiquitin ligase Deltex, which is known to play a role in ligand-independent intracellular activation of Notch by promoting its mono-ubiquitinated state (*Hori et al., 2011*). We found that punctate cytoplasmic Deltex expression increased significantly in AMPs expressing InRCA compared to the control (*Figure 6A–C* and *Figure 6—source data 1*). Consistently with this observation, targeted expression of Deltex in AMPs or attenuation of its repressor Supressor of Deltex (Su (Dx)) both led to an overproliferation phenotype (*Figure 6D,E,I* and *Figure 6—source data 1*). *Hori et al. (2011)* proposed that non-visual β-arrestin homolog Kurtz (Krz) binds together with Deltex to the Notch receptor, leading to its poly-ubiquitination and subsequent degradation. We tested Krz function in AMPs and found that both increasing and decreasing Kurtz levels leads to overproliferation of AMPs (*Figure 6—figure supplement 2* and *Figure 6—source data 1*). This suggested that the stoichiometry of Krz and Dx levels regulates Notch activation in

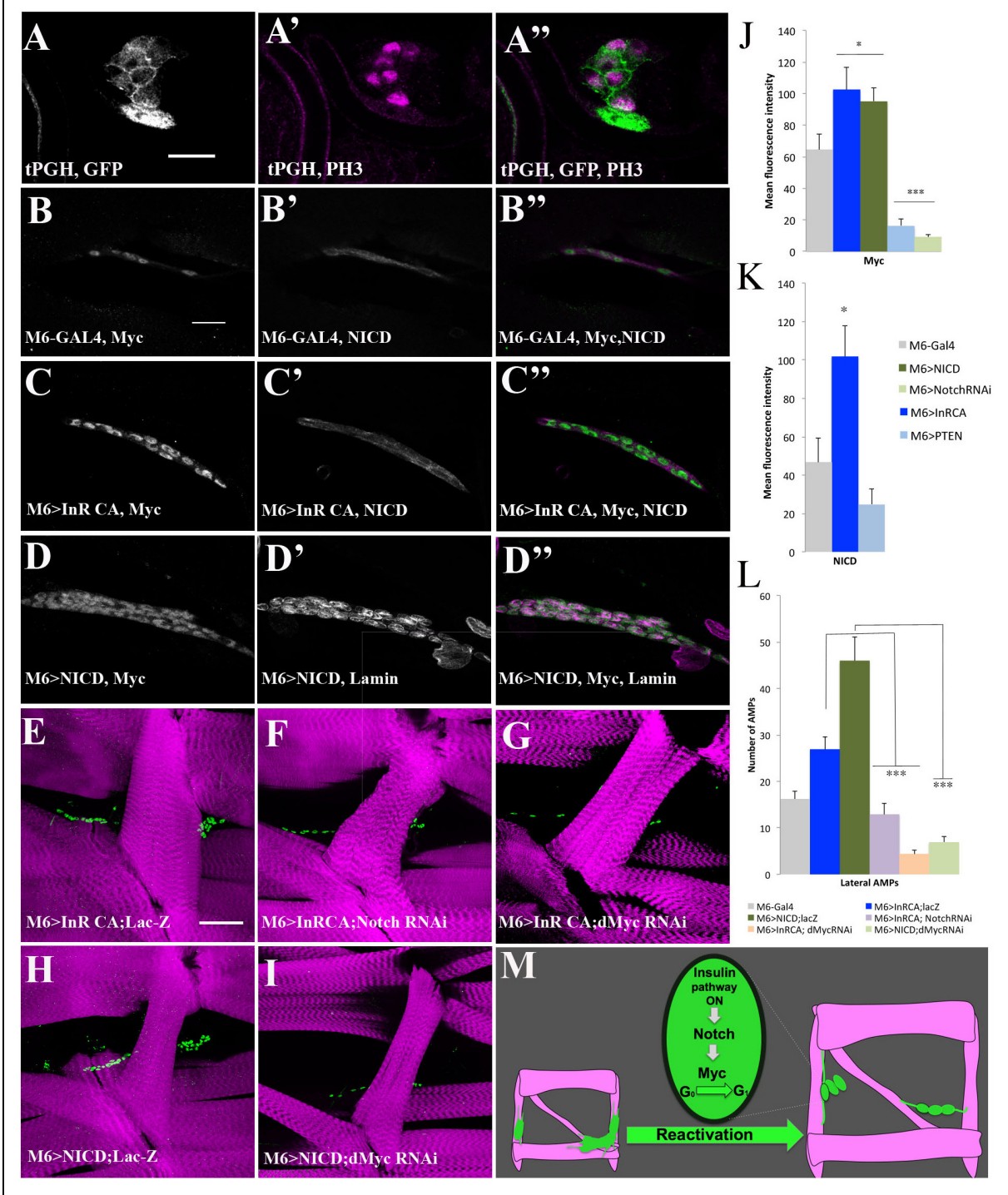

**Figure 5.** Myc acts downstream of Insulin and Notch pathways during AMP reactivation. (**A, A"**) A single cluster of AMPs from the tPGH third-instar larvae stained for GFP to reveal activation of PI3K/Insulin pathway and for phospho-histone H3 (PH3) to identify AMPs that undergo proliferation. Note that PH-GFP localizes to the cell membranes, indicating the activity of PI3K/Insulin signaling in AMPs that proliferate. (**B–D"**) Single clusters of third-instar larva lateral AMPs stained for dMyc and NICD (**B–C"**) and for dMyc and Lamin (**D–D''**). (**B, B"**) Control m6-GAL4 larva. (**C, C"**) m6-GAL4-driven expression of Inr-CAAX in AMPs upregulates dMyc and NICD expression. (**D, D"**) Targeted expression of NICD in AMPs results in an increased dMyc signal in AMPs. (**E–I**) Double transgenic mutant contexts and their effects on number of lateral AMPs. Attenuations of Notch (**F**) and dMyc (**G**) rescue the InRCA-induced overproliferation phenotype. Similarly, attenuating dMyc in AMPs expressing NICD dramatically reduces AMP numbers (**I**) compared to NICD context (**H**). (**J**) Mean fluorescence intensity of the dMyc signal detected in loss- and gain-of-function contexts for Insulin and Notch pathway components. (**K**) Mean fluorescence intensity of the NICD signal detected in InRCA and PTEN contexts. (**L**) Mean number of lateral AMPs counted in

*Figure 5 continued on next page*

Figure 5 continued

different genetic contexts shown in (E–I). (***) and (**) indicate $P \leq$ 0.001 and $P \leq$ 0.01, respectively. Scale bars are (A, A"): 9 microns; (B–D"): 15 microns; (E–I): 45 microns. (M) Schematic illustration of genetic hierarchy between Insulin, Notch and Myc during AMP reactivation.

The following source data and figure supplement are available for figure 5:

**Source data 1.** Table showing mean number of dorsal, lateral and ventral AMPs in the abdominal segments from the genotypes shown in *Figure 5E–I, L* and *Figure 5—figure supplement 1*.

**Figure supplement 1.** Proliferation of AMPs is positively regulated by Insulin, Notch and their downstream target Myc.

AMPs (see schemes in *Figure 6—figure supplement 2*). Indeed, reducing Deltex by RNAi-based attenuation or by overexpressing Su (Dx) led to an increased number of AMPs (*Figure 6—figure supplement 2* and *Figure 6-source data 1*) similar to that observed in the Deltex overexpression context (*Figure 6D,I* and *Figure 6—source data 1*), whereas simultaneous overexpression of Deltex and Kurtz had no effect on AMP numbers (*Figure 6—figure supplement 2* and *Figure 6—source data 1*). It has also been reported that Shrub, a component of the ESCRT-III complex that promotes Notch degradation in multivesicular bodies (MVBs), acts as a negative regulator of ligand-independent Notch activity (*Hori et al., 2011*). We thus tested whether Shrub attenuation could promote AMP proliferation. We found that a downregulation of Shrub in AMP cells results in a burst of AMP proliferation (*Figure 6F,I* and *Figure 6—source data 1*). Consistent with all these data, overexpressing Deltex or Kurtz in InRCA context increased AMP cell numbers compared to InRCA alone (*Figure 6G,I*, *Figure 6—figure supplement 1* and *Figure 6—source data 1*). On the other hand, overexpressing Deltex in AMPs in which the Insulin pathway was attenuated by PTEN restored AMP proliferation back up to wild-type levels (*Figure 6H,I* and *Figure 6—source data 1*). Taken together, this body of evidence suggests that during AMP reactivation, the Insulin pathway activates Notch in a Deltex and Shrub-involving ligand-independent way.

## Muscle niche-derived dIlp6 reactivates AMPs

The specific role of the Insulin pathway in the proliferation of AMPs suggested it was induced by locally-secreted Insulin receptor ligands, the Insulin-like peptides (dIlps). We first tested mutants for *dIlp2, dIlp5* and *dIlp6* (*Grönke et al., 2010*) and found that AMP proliferation was only inhibited in a *dIlp6* loss-of-function background (*Figure 7A–C,J*, *Figure 7—figure supplement 1* and *Figure 7—source data 1*). To identify the source of the dIlp6, we expressed a dominant-negative (DN) form of *Drosophila* dynamin protein called Shibire (Shi-DN) in larval muscles, neural cells and glial cells (*Figure 7D–F,J*, *Figure 7—figure supplement 1* and *Figure 7—source data 1*). Shi-DN affects vesicular trafficking and thus inhibits the secretion of signaling molecules from the cell (*Seugnet et al., 1997*). We found that blocking the secretion from larval muscles affected the proliferation of AMPs and reduced their numbers (*Figure 7D*, *Figure 7—figure supplement 1* and *Figure 7—source data 1*), whereas no effect was observed when Shi-DN was expressed in either neural cells (*Figure 7E,J*, *Figure 7—figure supplement 1* and *Figure 7—source data 1*) or glial cells (*Figure 7F,J*, *Figure 7—figure supplement 1* and *Figure 7—source data 1*). A key role of muscles in producing dIlp6 and inducing AMP proliferation is further supported by the reduction of AMP numbers in larvae with attenuated *dIlp6* expression in muscles but not in glial cells (*Figure 7G,I,J*, *Figure 7—figure supplement 1* and *Figure 7—source data 1*). Conversely, an increased number of AMPs was detected in muscle-specific overexpression of dIlp6 (*Figure 7H,J*, *Figure 7—figure supplement 1* and *Figure 7—source data 1*). To understand the link between persisting cellular extensions and reactivation of AMPs, we attempted to modulate filopodia formation by attenuating the *DAAM* gene encoding one of formins known to be involved in filopodia dynamics at axon growth cones (*Gonçalves-Pimentel et al., 2011*). We found that *DAAM*-attenuation leads to an altered AMP proliferation in third-instar larvae (*Figure 7M* and *Figure 7—source data 1*) that correlates with reduced length of cellular protrusions observed in *DAAM*-RNAi second-instar larvae (*Figure 7L*, compare to wild-type shown in *Figure 7K*).

Taken together, the data suggest that muscle behaves as an AMP niche and plays a driving role in AMP reactivation in later larval life (see AMP reactivation scheme, *Figure 8*).

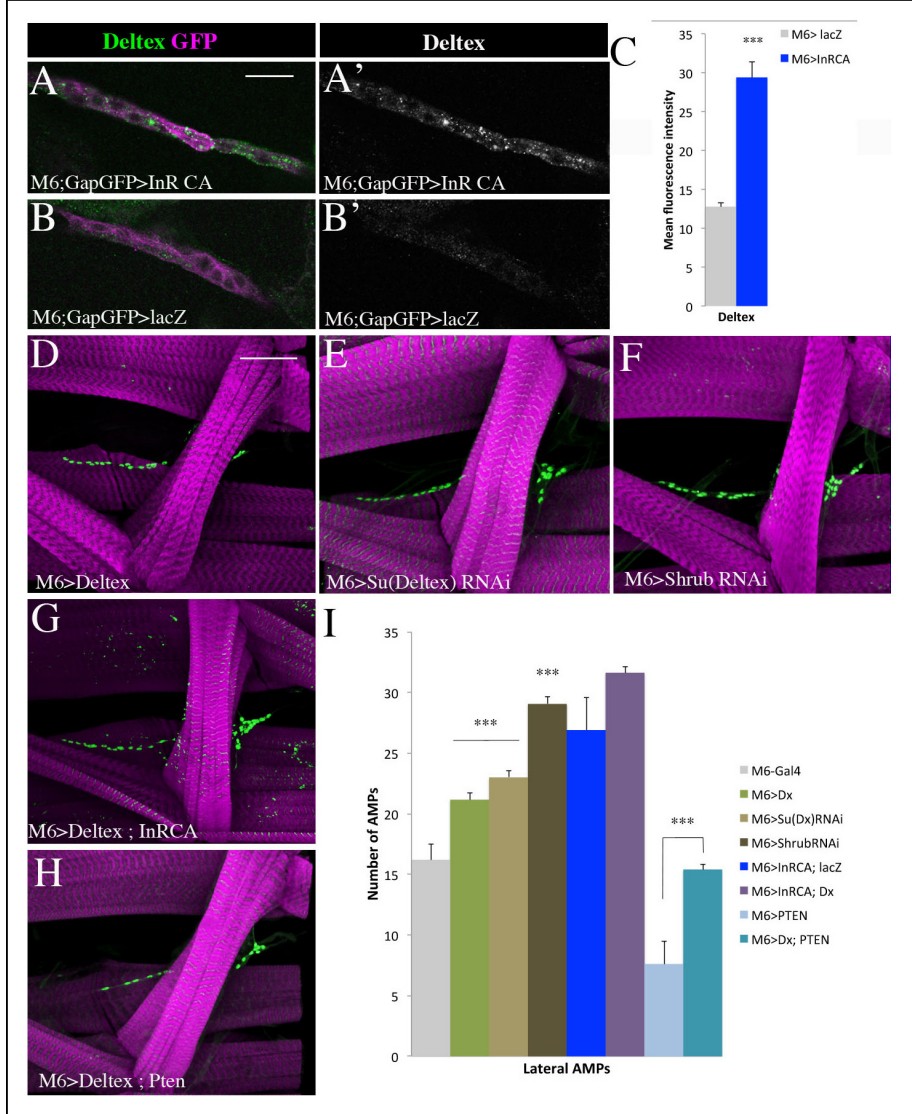

**Figure 6.** Insulin-driven Notch activation in AMPs involves Deltex. (A-B') Single clusters of third-instar larva lateral AMPs stained for Deltex and GFP. (A–A') There is greater punctate Deltex expression in AMPs expressing constitutively activated InR than in control larva (B-B') expressing *lacZ*. (C) Mean fluorescence intensity of the Deltex signal detected in gain-of-function context for Insulin *versus* wild-type. (D-F) Components of ligand-independent Notch activation have impacts on AMP cell numbers. AMP-targeted expression of Deltex (D), attenuation of Su (Deltex) (E) or attenuation of Shrub (F) all lead to an AMP overproliferation phenotype. The key role of Deltex as an activator of AMP proliferation is confirmed by an increased number of AMPs in embryos with M6-targeted expression of InRCA and Deltex (G) and further supported by partial rescue of AMP number when co-expressing Deltex with the PTEN Insulin pathway inhibitor (H). (I) Graphical representations of mean number of lateral AMPs in genetic contexts shown in (D-H). (***) indicates $P \leq 0.001$. Scale bars are (A, B'): 15 microns; (D–H): 45 microns.

The following source data and figure supplements are available for figure 6:

**Source data 1.** Table showing mean number of AMPs in the abdominal segments from the genotypes shown in *Figure 6D–I* and *Figure 6—figure supplement 1* and *2*.

**Figure supplement 1.** Ligand independent activation of Notch promotes proliferation of AMPs.

**Figure supplement 2.** Role of Kurtz and Deltex in reactivation of AMPs.

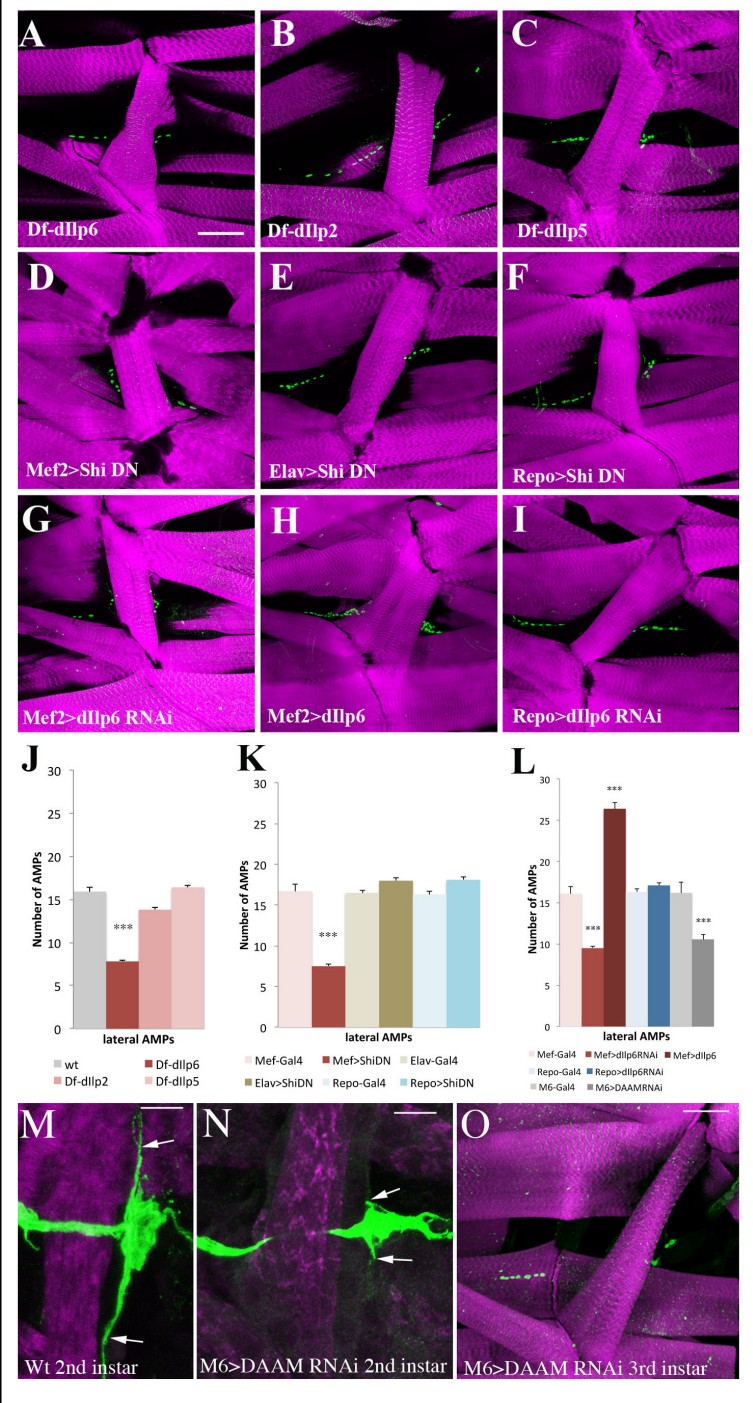

**Figure 7.** Larval muscles regulate AMP proliferation via Insulin-like peptide dIlp6. (**A–I**) Flat preparations of the mid-stage matched third instar larvae stained for Twist (green) labeling AMP nuclei and Phalloidin (magenta) labeling larval muscles. One abdominal lateral group of AMPs is shown. (**A**) Larvae mutant for *dIlp6 (Df-Ilp6)* shows a lower AMP count. (**B, C**) No changes in AMP number are observed in *dIlp2* or *dIlp5* mutant larvae. (**D**) Muscle-targeted expression of the dominant-negative form of *shibire (DN-shi)* leads to a decrease in AMP cell number. (**E, F**) Elav-Gal4-driven expression of *DN-shi* in neural cells or Repo-Gal4-driven expression in glial cells have no effects on AMP number. (**G**) Attenuation of *dIlp6* in larval muscle leads to a decrease in AMP number while (**H**) muscle-specific gain-of-function of *dIlp6* leads to an increase in AMP number. (**I**) No change in AMP number is observed after RNAi-based attenuation of *dIlp6* in glial cells. (**J**) Mean number of lateral AMPs counted in different genetic contexts shown in A–I and O. (***) indicates $P \leq 0.001$. (**K**) Posterior lateral AMP revealed by GFP staining (green) of M6-gapGFP second-instar larvae. Arrows indicate long AMP filopodia extending along the segment border muscle (Phalloidin staining, in magenta). (**L**) A similar view of posterior lateral AMP from second instar DAAM-RNAi larvae. Arrows point to short filopodia. (**M**) Reduced AMP numbers in third instar larvae induced by M6-targeted attenuation of DAAM. Scale bar in (**A–I**) and (**O**): 45 microns; in (**M, N**): 25 microns.

*Figure 7 continued on next page*

*Figure 7 continued*

The following source data and figure supplement are available for figure 7:

**Source data 1.** Table showing mean number of dorsal, lateral and ventral AMPs in the abdominal segments from the genotypes shown in *Figure 7A–L*
and *Figure 7—figure supplement 1*.
**Figure supplement 1.** Muscle released dIlp6 is required for the activation of dorsal and lateral but not ventral AMPs.

## Discussion

How the stem cells specified during development keep their non-differentiated quiescent properties and how they are reactivated from their dormant state remain central challenges in stem cell biology. Here, we applied a *Drosophila* model to analyze cellular and molecular events underpinning Adult Muscle Precursor (AMP) reactivation from the dormant state, and identified a muscle niche-driven Insulin-Notch-dMyc regulatory cascade (*Figure 8*) that governs AMP entry into proliferation.

### Homing behavior of AMPs to their muscle niche

It is widely accepted that stem cells reside in a specific microenvironment, the niche, defined as an interactive structural unit organized to facilitate cell-fate decisions in a proper spatiotemporal manner (*Moore and Lemischka, 2006*). The niche of satellite cells located underneath the basal lamina of myofibers is composed of cellular and extracellular components (muscle fiber and basal lamina) that are sufficient for satellite cell activation, proliferation and differentiation (*Zammit et al., 2004*). Niche properties, particularly of its muscle component, also appear crucial for the regenerative potential of satellite cells during aging (*Chakkalakal et al., 2012*) and satellite cell engraftment in cell therapy approaches (*Boldrin et al., 2012*).

The stem-like *Drosophila* AMPs specified in mid-embryogenesis lie at the origin of all the adult *Drosophila* muscles (*Bate et al., 1991*), but the lack of appropriate genetic tools means we still know little of their behavior and niche requirements. Here, we monitored the morphology of embryonic AMPs using a m6-gapGFP sensor line and found that in addition to the long cytoplasmic extensions that connect AMPs together (*Figeac et al., 2010*), they also send out numerous thin filopodia and display homing behavior to a set of surrounding muscles. The newly specified AMPs initially display spherical shapes with short thin filopodia distributed around their surface, but shortly afterwards they start to send filopodia in a more directional way, become more elongated extending along the neighboring muscles and connecting them via stabilized cellular protrusions. Thus, the dormant AMPs, like vertebrate satellite cells, adapt their shapes to muscle niche and became tightly associated with muscle fibers, an assumption supported by dotty integrin expression associated with both AMP cell bodies and their muscle-connecting filopodia. Interestingly, the quiescent satellite cells also have the capacity to produce cellular extensions, called nanotubes, connecting muscle stem cells to the muscle fiber (*Tavi et al., 2010*), which further argues that AMPs and satellite cells display similar behavior. How the homing of satellite cells and AMPs is regulated remains to be explored, but the Notch pathway, which is activated in AMPs and required to produce basal lamina by adhering to muscle satellite cells (*Bröhl et al., 2012*), appears to play a central role. It is also unknown how satellite cells behave during their reactivation, how they adapt their shapes, and when they lose nanotube connections to muscle niche. Using membrane-targeted GFP enabled us to follow the behavior of AMPs and, for the first time, to visualize them at the time they are reactivated from the dormant state. Our data reveal that the AMPs maintain their elongated shapes with filopodia extending along the muscle niche during the first events of proliferation, which supports the view that muscles play an instructive role during AMP reactivation.

### Insulin-Notch-dMyc cascade controls AMP reactivation and proliferation

AMPs are reactivated from their quiescent state at the mid-second larval instar, but the signals and intrinsic molecular mechanisms regulating their entry into proliferation remain unknown. It has previously been shown that the Insulin/TOR signaling pathway controls the exit of *Drosophila* neural stem cells from their dormant state (*Chell and Brand, 2010*; *Sousa-Nunes et al., 2011*). Global gene

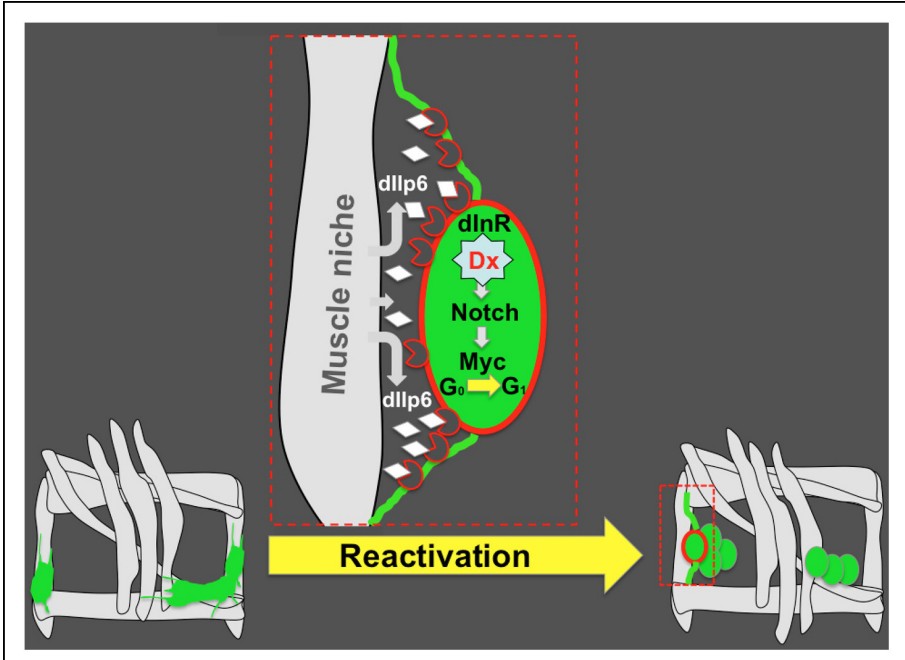

**Figure 8.** Niche role of muscle in AMP reactivation. Scheme illustrating the muscle niche-induced Insulin/Notch/dMyc cascade governing the reactivation of dormant AMPs. During embryonic stages, quiescent AMPs send out filopodia and make contact with neighboring muscles. These AMP-to-muscle ties persist until the AMPs are reactivated at mid-second larval instar, facilitating the reception of the inductive dllp6 signal emitted by the muscle niche. In reactivated AMP (depicted in red), activation of the Insulin pathway leads to a Deltex-involving activation of Notch and induces AMP proliferation through the Notch target Myc.

expression profiling also revealed that Insulin Growth Factor (IGF) signaling components are upregulated in activated satellite cells (*Pallafacchina et al., 2010*). We therefore tested whether the Insulin pathway could reactivate *Drosophila* AMPs. We found that Insulin and TOR signaling components effectively promote the exit of AMPs from their dormant state, while also positively regulating AMP proliferation. However, increased Insulin/TOR signaling alone is not sufficient to end AMP quiescence during embryonic stages, suggesting a more complex regulatory cascade for the control of AMP reactivation. It has been demonstrated that the reactivation of dormant neuroblasts is driven by a nutritional checkpoint during the second larval instar (*Chell and Brand, 2010*; *Sousa-Nunes et al., 2011*). Given that AMPs also remain quiescent in nutrient deprivation settings, we hypothesize that a nutrient-dependent switch in metabolism may contribute to AMP reactivation, and might thus be responsible for the inability of embryonic AMPs to enter proliferation. Quiescent stem cells have few mitochondria and use anaerobic metabolism, whereas activated cells switch their metabolism to a high ATP energy output aerobic glycolysis in order to support the high-level macromolecular synthesis required for proliferation, (*Montarras et al., 2013*). Interestingly, genes encoding glycolytic and pyruvate metabolic enzymes known to be downstream of Insulin (*Tixier et al., 2013*) act as upstream regulators of the Notch pathway (*Saj et al., 2010*), providing a potential link between metabolism and Notch in regulating the quiescent versus activated state of stem cells. A well-known feature of quiescent non-differentiated cells is that they keep Notch activated (*Bjornson et al., 2012*), but Notch receptor activation can also trigger proliferation (*Baonza and Garcia-Bellido, 2000*), including the proliferation of satellite cells (*Qin et al., 2013*). Here, we followed the GFP expression driven by a Notch-responsive element of the *m6* gene and found that dormant AMPs are GFP-positive, which suggests that, like in satellite cells, Notch is involved in setting the AMP quiescent state. On the other hand, we also observed that during larval stages Notch promotes AMP proliferation, indicating that Notch might play a dual role, again like in satellite cells (*Qin et al., 2013*). Notch acting in concert with Numb is also known to control asymmetric divisions of adult satellite cells (*Kuang et al., 2007*). However, in *Drosophila*, Numb has no impact on AMP proliferation, indicating a novel role for Notch in promoting symmetric cell divisions of AMPs. As

Notch regulates the proliferation of cancer cells via Myc (*Palomero et al., 2006*; *Weng et al., 2006*; *Liao et al., 2007*; *Ishikawa et al., 2013*), and Myc promotes the proliferation of myoblasts during development and regeneration (*Li et al., 2012*), we tested whether dMyc was involved in AMP reactivation in *Drosophila*. Our data show that dMyc is indeed required for AMP reactivation and proliferation. Genetic rescue experiments show that Notch acts downstream of the Insulin pathway and dMyc functions downstream of Notch. Interestingly, it has been shown that coordinated activation of Insulin and Notch pathways also regulates the self-renewal and differentiation of intestinal *Drosophila* stem cells (*Foronda et al., 2014*). Here, our data suggest that Notch pathway activation downstream of Insulin involves Deltex and is likely ligand-independent (*Figure 7*). The role of ligand-independent Notch activation, which involves interaction with Hif-α, has been reported as essential for normal *Drosophila* blood cell development (*Mukherjee et al., 2011*). Our findings further highlight the biological importance of non-canonical Notch activation and provide early evidence for its role during stem cell-niche interactions.

## Muscle niche reactivates AMPs via dIlp6

The key role of the Insulin pathway in AMP proliferation raised an obvious question as to the source and identity of the Insulin Receptor (InR) ligand initiating AMP reactivation. In *Drosophila*, three Insulin-like peptides (dIlp2, dIlp3 and dIlp5) secreted to the hemolymph by mNSC cells control the systemic growth of the organism (*Brogiolo et al., 2001*), whereas dIlp6 produced by the glial cells acts locally and promotes the proliferation of the neuroblasts (*Sousa-Nunes et al., 2011*).

Here we show that dIlp6 is also crucial for reactivating AMPs from their dormant state. On the other hand, the lack of substantial effect on AMP proliferation observed in *dIlp2, dIlp3* and *dIlp5* mutants suggests that dIlps secreted to the hemolymph are unable to reactivate AMPs and that direct contact between AMPs and surrounding muscles via long cellular protrusions promotes the reception of local, muscle-derived dIlp6 by the closely-associated AMPs.

Thus, like for satellite cells (*Chakkalakal et al., 2012*), muscle appears to play a niche role while also proving essential to the activation of quiescent AMPs (*Figure 8*). Whether the Insulin/IGF signals play a conserved role in this process remains an open question, but the finding that numerous IGF signaling components are upregulated in activated satellite cells (*Pallafacchina et al., 2010*) argues for this possibility.

Taken together, our data reveal several new features of the transient *Drosophila* muscle stem cells called AMPs, particularly their direct contact with muscles, which behave as an AMPs niche. Muscle appears to be the source of local inductive Insulin signals that reactivate AMPs and, via non-canonical Notch and its target dMyc, induce their proliferation in larval stages (*Figure 8*). If all the components of this cascade prove to be evolutionarily conserved, it is tempting to speculate that this same cascade may also control the reactivation of human satellite cells.

# Material and methods

## Fly stocks

All *Drosophila melanogaster* stocks were grown on standard medium at 25°C. The overexpression experiments were performed by UAS-GAL4 system (*Brand and Perrimon, 1993*). $w^{1118}$ was used as a wild type. The following strains were used: *UAS-PTEN, UAS-TSC1, TSC2* (*Potter et al., 2001*) *UAS-dMyc* (*Orian et al., 2007*) (gift from F. Demontis, Judes Children Hospital, USA), *Tubulin-PH-GFP (tPGH)* (gift from B. Edgar, University of Heidelberg, Germany), Duf-GAL4 (gift from K. Vijayraghavan, NCBS, India), *UAS-NotchDN, UAS-NICD, UAS-Deltex UAS-Kurtz, UAS-Su (Dx)* (gift from Spyros Artavanis-Tsakonas, Harvard Medical School, USA), *UAS-Shrub* (a gift from Fen Biao-Gao, University of Massachusetts Medical School, USA), *UAS-ShrubRNAi* (gift from Janice A. Fischer, Texas University, USA),

*Mef-Gal4, UAS-Krz, UAS-LacZ, UAS-InRCA, UAS-RHEB, UAS-mCD8Cherry, UAS-ShibireDN, UAS-rpr, Elav-GAL4, Repo-GAL4* and the mutant stocks for *Df (Ilp6), Df (Ilp2), Df (Ilp5)* were obtained from the Bloomington stock center (BL27390, BL27889, BL1776, BL8250, BL9689, BL27391, BL5811, BL5824, BL458, BL7415, BL30885, BL30881, BL30884, respectively). *UAS-KurtzRNAi, UAS-DeltaRNAi, UAS-SerrateRNAi, UAS-DeltexRNAi, UAS-Su (Dx)RNAi, UAS-dMycRNAi, UAS-NotchRNAi, UAS-Ilp6RNAi, UAS-shrubRNAi, UAS-DAAMRNAi* (V103756, V37287, V27174, V7795, V103814, V106066,

V10002, V102465, V106823, V24885, respectively) came from Vienna Drosophila Research Centre (VDRC). Double transgenic lines *UAS-InRCA;UAS-NotchRNAi, UAS-InRCA;UAS-dMycRNAi, UAS-InRCA;UAS-LacZ, UAS-NICD;UAS-dMycRNAi, UAS-InRCA;UAS-Krz, UAS-InRCA;UAS-Dx, m6-Gal4; UAS-Krz, m6-Gal4;UAS-Dx and UAS-PTEN;UAS-Dx* were generated by standard genetic crosses. Crosses and embryo collection were performed at 25°C.

## Generation of m6-gapGFP and sensor-driver stocks

To generate m6-gapGFP, the eGFP coding sequence from the pGreen Pelican vector (*Barolo et al., 2000*) was replaced by the gapGFP coding sequence (a fusion of the myristylization sequence from *GAP43* gene to GFP, designed to target the GFP to cell membrane) from the pCA-gapGFP vector (gift from A. Chiba, University of Miami, USA). The regulatory sequences from the upstream region of the *m6* gene, with expression seen in AMPs (*Figeac et al., 2010*), were PCR-amplified from the genomic DNA and inserted into the multiple cloning site (MCS) region. Germline transformation of the m6-gapGFP vector and the generation of transgenic lines were performed by the Fly-Facility platform (www.fly-facility.com) using the standard P-element-based transgenesis method. Sensor-driver stocks consisting of m6-gapGFP and m6-GAL4 (targeting AMPs) or Elav-GAL4 (targeting neural cells) were generated by standard genetic crosses.

## Histology, AMP cell counting and statistical analyses

Embryos from the synchronized cages were collected within 2 h. L1 larvae collected within 2 h of hatching were grown on standard medium at 25°C. At 96 h AEL period, mid-third instar larvae were pinned flat and dissected in calcium-free PBS. Larvae staging was supported by the mouth hook morphology. The internal organs were removed to expose the body-wall muscles, and fixed in 4% formaldehyde for 20 min. After fixation, the dissected larvae were washed twice in PBS and used for immunostaining via standard procedures. AMPs were visualized by staining with Twist antibody, and the number of AMPs per group was counted in the A2–A5 abdominal segments. All samples were co-stained with phalloidin, and only larvae with intact muscles were selected for AMP counting and quantification.

Flat preparations of the first and second instar larvae were prepared according to *Marley and Baines (2011)*, and used for the immunostaining as stated above.

For each genotype, at least eight larvae were dissected and the ventral, lateral and dorsal AMP groups were used for counting (sample sizes for each genotype are indicated in *Figure 4—source data 1*, *Figure 5—source data 1*, *Figure 6—source data 1* and *Figure 7—source data 1*). For each experiment, we calculated a mean value that was used to generate the graphs. The standard error of the mean (SEM) was applied to calculate the error bars. A student's t-test and Prism software were used to calculate the *P*-values.

## Growing the larvae in nutrient-restricted conditions

Embryos from the synchronized cages were collected 2 hr after laying and allowed to develop on plates with sugar-agar medium (5% sucrose, 1% agar) at 25°C. L1 larvae were collected and allowed to grow on fresh sugar-agar plates at 25°C. At 96 h AEL, the larvae were pinned flat and dissected in calcium free PBS. The internal organs were removed to expose the body-wall muscles, and mounted on a coverslip to visualize AMPs under the confocal microscope.

## Genetic epistasis experiments

Genetic epistasis experiments were performed to determine interactions between Insulin and Notch pathways and between Notch and dMyc during AMP reactivation. We first generated double UAS-InRCA; UAS-NotchRNAi, UAS-Notch-Intra; UAS-MycRNAi, UAS-InRCA; UAS-Deltex, UAS-PTEN; UAS-Deltex, UAS-InRCA; UAS-Kurtz; UAS-Deltex; UAS-Kurtz and UAS-InRCA; UAS-lacZ transgenic lines. Each of these double transgenics was then crossed with the m6-GAL4 driver. The derived synchronized mid-stage third instar larvae were dissected and immunostained to reveal muscles and AMPs. AMPs were counted and their numbers compared against m6>InRCA and m6>Notch-Intra contexts.

## Antibody staining and imaging

Fluorescent staining was performed using the following antibodies: rabbit anti-Twi (*Figeac et al., 2010*) (1:300), rabbit anti-dMyc (1:300) (Santa-Cruz Biotechnology), goat anti-GFP (1:1000) (Biogenesis), rabbit anti-PH3 (1:1000) (Millipore), mouse anti-NICD (1:150), rat anti-Deltex (1:50) (kindly provided by S. Artavanis-Tsakonas, Harvard Medical School, USA), mouse anti-Lamin (1:1000) (DHSB LC28.26), rat anti-Tropomyosin (1–200; Babraham Bioscience Technologies, UK; BT-GB-141), mouse anti-αPS1 (1:50; DHSB; DK.1A4), mouse anti-βPS (DSHB CF.6G11), Phalloidin-TRITC (1:1000) (Sigma). Cy3, Cy5 and Alexa 488-conjugated secondary antibodies (Jackson ImmunoResearch) were used (1:300). Embryos were mounted in Fluoromount-G anti-fade reagent (Southern Biotech). Labeled embryos were analyzed using Leica SP5 and SP8 confocal microscopes. 3D reconstructions of the images were generated using Imaris software (Bitplane).

## Signal intensity measurements and statistical analyses

All confocal images used to measure signal intensity were acquired at the same microscope settings. Equal numbers of stacks per image were taken for the different genetic contexts, and mean fluorescent intensity of a single cluster of AMPs from the abdominal segments of the mid-third instar larvae was measured using ImageJ software. For each genotype, 4-6 larvae were dissected and 12–15 segments were analysed. Mean fluorescence intensity for a given cluster of AMPs was determined by averaging the signal intensities measured in three representative AMP cells from that cluster. Statistical analyses were performed by a Student's *t*-test using Prism software and Microsoft Excel.

## Acknowledgements

This work was supported by ANR grants "MYO-ID" and "ID-CELL-SPE", a TEFOR Infrastructure grant, the "Equipe FRM" grant, and the AFM grants to KJ. The authors thank the Bloomington, VDRC and DGRC Resource Centres for providing reagents and fly stocks.

## Additional information

### Funding

| Funder | Grant reference number | Author |
| --- | --- | --- |
| Fondation pour la Recherche Médicale | Equipe grant DEQ20140329515 | Krzysztof Jagla |
| Agence Nationale de la Recherche | Myo-ID grant ANR-09-BLAN-0279-01 | Krzysztof Jagla |
| Agence Nationale de la Recherche | ID-CELL-SPE grant ANR-12-BSV2-0017-01 | Krzysztof Jagla |
| Association Francaise contre les Myopathies | AFM-15865 | Krzysztof Jagla |

The funders had no role in study design, data collection and interpretation, or the decision to submit the work for publication.

### Author contributions

RA, Acquisition of data, Analysis and interpretation of data, Drafting or revising the article; MZ, Acquisition of data, Analysis and interpretation of data; JPDP, This author contributed by setting genetic crosses, preparing biological samples for analyses and providing excellent technical assistance, Acquisition of data; KJ, Conception and design, Analysis and interpretation of data, Drafting or revising the article

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
