## [Decision Letter]

Thank you for submitting your work entitled "Muscle niche-driven Insulin-Notch-Myc cascade reactivates dormant Adult Muscle Precursors in *Drosophila*" for peer review at *eLife*. Your submission has been favorably evaluated by K VijayRaghavan (Senior Editor), a Reviewing Editor, and three reviewers.

The reviewers have discussed the reviews with one another and the Reviewing Editor has drafted this decision to help you prepare a revised submission.

The reviewers appreciate this original study of the communication between *Drosophila* Adult Muscle Precursors (AMPs) and their *niche*, providing interesting parallels with the niche of satellite cells in adult vertebrates. The authors report observations on filopodia produced by AMPs to contact the muscle and investigate signaling pathways that control AMP behaviour, with novel findings on ligand independent activation of Notch by upstream action of the insulin receptor, which in turn is controlled by dIlp6 produced by the muscle.

The following points need to be addressed:

1) Niche 'homing' behavior of the AMPs is the least well-supported part of the manuscript and relies largely on description of AMP position and morphology. Are the preferences for filipodial contact between AMPs and specific muscle fibers relevant, or merely due to proximity? This is easily tested using ablation of specific fibers or genetic backgrounds in which they fail to form (the Keshishian lab did this to great effect for studies of motorneuron innervation in the mid-1990s, for example). Such studies would bolster the idea that the cells home to the niche and may reveal some interesting insights.

The second and third paragraphs of the subsection “AMPs display homing behavior and became tightly associated with neighboring muscles” (“Embryonic AMPs are the immediate neighbors of somatic muscles […] tightly associated”; “Interestingly, filopodial AMP-muscle connections appear to follow a stereotyped pattern […] are decorated by punctate integrin’s expression”) need to be rephrased, as the description is not clear. Because filopodial extensions are generally viewed as highly dynamic, the characterization of AMP filopodia should include: range, dynamics (stability); target specificity; position of integrin accumulation. This is important to support the conclusion that "direct contacts between AMPs and surrounding muscles favor reception of local muscle-derived dIlp6" (paragraph two, subheading “Muscle niche reactivates AMPs via dIlp6”). If technically possible, an improved 3-D view (reconstruction?) of AMP/Muscle contacts via filopodia would help non-expert readers to appreciate the positioning of AMPS relative to internal, median and external muscle *layers*. What happens to signaling if filipodia are disrupted?

In conclusion, the authors do little to investigate the relevance of the filipodia in either homing or signaling. A detailed analysis is probably beyond the scope of this article but some attempt to clarify ideas about their function should be made, along the lines indicated above.

2) A second point concerns the Gal4 drivers used.

The authors’ statement that muscles are the source of Dilp6 acting on AMPs, is based on using a pan-mesodermal Mef2 driver (Figure 6). This driver could also affect the AMPS. An additional specific muscle driver e.g. MHC-Gal4 should be used. Furthermore, since drivers exist that are only active in subsets of somatic muscles, some contacted by AMPs, others not, use of these would be informative as to whether direct contacts between specific muscles and AMPs are instructive.

vM6-Gal4 is generated from a Notch dependent enhancer. This complicates the experiments when Notch is manipulated. For example, inhibition of Notch would reduce expression of M6-Gal4. This can in turn impact on expression of the other RNAi's and modify the phenotypes. At least some experiments should be repeated with an independent Gal4 to verify the effects.

3) The authors argue that the effects of Notch involve ligand-independent signaling, but they have not really tested that. Dx and Su(Dx) can also affect ligand dependent signaling. A strategy that blocks ligand dependent signaling would clarify this issue e.g. by blocking the ADAM protease.

The conclusion that N acts downstream of InR in stimulating AMP proliferation is novel and of potential strong impact in the field. Yet, the authors should leave open the possibility that N acts also independent of InR since overexpressing NICD induces higher number of AMPs (Figure 3 and Figure 4) than InRCA, although not higher Myc levels (Figure 4).

[Editors' note: further revisions were requested prior to acceptance, as described below.]

Thank you for resubmitting your work entitled "Muscle niche-driven Insulin-Notch-Myc cascade reactivates dormant Adult Muscle Precursors in *Drosophila*" for further consideration at *eLife*. Your revised article has been favorably evaluated by K VijayRaghavan (Senior Editor), a Reviewing Editor, and three reviewers. The manuscript has been greatly improved and the new Video 4 was appreciated, but there are some remaining modifications that need to be made to the manuscript before acceptance, as outlined below:

1) On the one hand, the authors use the expression from the m6 enhancer (m6-GFP) expression to argue that Notch is active in the AMPs, but then they now argue that the expression from the same enhancer (in m6-Gal4) can't be reduced when Notch activity is attenuated. These claims are rather contradictory. Most likely there is a simple explanation (perdurance of Gal4 and GFP in the latter?) but this does make the results confusing for the reader (e.g. subheading “Notch acts downstream of the Insulin pathway and regulates the proliferation of AMPs via dMyc” and in the Discussion: "Here we followed the GFP expression driven by a Notch-responsive element of the *m6* gene and found that the Notch pathway was active in dormant AMPs"). Can they be confident that this is reading out Notch activity at that time? Maybe the enhancer requires Notch for activation but not for maintenance? Given the data they have provided, I suggest that they tone down their statements.

2) Teasing apart the ligand dependent versus ligand independent Notch activity is tricky. The authors discuss in the rebuttal their experiment with the KuzDN, which gives minor effects that they argue should be ignored. But at the same time they place quite a strong emphasis on the role of Deltex despite the fact that the results in this section are confusing and contradictory (e.g. over-expressing and attenuating Deltex have the same effects, likewise Su(dx), likewise kurtz). Given these contrary data, it doesn't seem appropriate to make major conclusions about the roles of these factors so I suggest that they remove the scheme from the right hand part of Figure 8 and further moderate their statements about the role of Dx. Altering Deltex activity does influence the AMPS but it is not clear that it is the main effector of the signaling through InR nor the critical regulator of Notch in this context.

3) Figure 6—figure supplement 1 presents a hypothetical scheme not directly connected to experiments described in the manuscript. It should be removed.

---

## [Author Response]

*The following points need to be addressed:*

1) Niche 'homing' behavior of the AMPs is the least well-supported part of the manuscript and relies largely on description of AMP position and morphology. Are the preferences for filipodial contact between AMPs and specific muscle fibers relevant, or merely due to proximity? This is easily tested using ablation of specific fibers or genetic backgrounds in which they fail to form (the Keshishian lab did this to great effect for studies of motorneuron innervation in the mid-1990s, for example). Such studies would bolster the idea that the cells home to the niche and may reveal some interesting insights.

The AMPs interact with immediately neighboring muscles but seem to have some preferences in interactions so that for example the most anterior lateral AMP interacts preferentially with LT and LO1 lateral muscles whereas posterior lateral AMP is mainly connected to the SBM. As suggested in comment 1 we performed a series of targeted muscle ablations to test AMPs behavior in the context neighboring muscles are absent. To do so we expressed inducer of apoptosis Reaper specifically in muscle founder cells using Duf-Gal4 driver (we favor genetic ablation method over laser ablation). This has led to a range of muscle ablation phenotypes (new Figure 2) and the following observations:

1) In segments with partial loss of LT/LO1 muscles the anterior lateral AMP, which normally extends anteriorly to interact with LO1, LT3/2/1 (white arrow in Figure 2) remained tightly associated with posterior lateral AMP and interacted mainly with SBM muscle (white arrows in Figure 2).

2) In segments with loss of dorsal and dorso-lateral muscles and with some LT muscles persisting (Figure 2) the dorso-lateral AMPs interacted with remaining LT1/2 muscles (arrowhead in Figure 2) to which they do not connect in the wild type context (arrowhead in Figure 2) indicating some plasticity in AMPs connections.

3) In segments with a pronounced loss of dorsolateral and lateral muscles (Figure 2) the dorsal and dorso-lateral AMPs adopted rounded shapes (yellow arrows) and were unable to migrate to other segments or to the ventral region in which muscles were still present.

4) In embryos with a total muscle ablation the majority of remaining AMPs adopted rounded shapes (yellow arrows in Figure 2). The number of AMPs detected was drastically reduced (asterisks indicate lacking AMPs) suggesting that in the absence of muscles AMPs do not survive.

All these observations indicate that there is a plasticity of AMP interactions with neighboring muscles. However, there is also dorsal-ventral positional information restricting these interactions in a way dorsal AMPs are unable to migrate and interact with lateral muscles when dorsal muscles are ablated.

Also AMPs association with muscle niche appears crucial for their survival as AMPs number is drastically reduced when muscles are absent (see progressively increasing number of lacking AMPs, indicated by asterisks, correlating with the severity of muscle loss phenotypes – compare Figure 2).

After analyzing a range of muscle ablation phenotypes generated in Duf>Rpr context we assume that we already generated ablations similar to those that could be generated by drivers targeting subsets of muscles.

We also used anti-Elav staining to test whether in Duf>Rpr context the PNS is not affected (data not shown). We didn’t find any defects in the PNS.

*The second and third paragraphs of the subsection “AMPs display homing behavior and became tightly associated with neighboring muscles” (“Embryonic AMPs are the immediate neighbors of somatic muscles […] tightly associated”; “Interestingly, filopodial AMP-muscle connections appear to follow a stereotyped pattern […] are decorated by punctate integrin’s expression”) need to be rephrased, as the description is not clear. Because filopodial extensions are generally viewed as highly dynamic, the characterization of AMP filopodia should include: range, dynamics (stability); target specificity; position of integrin accumulation. This is important to support the conclusion that "direct contacts between AMPs and surrounding muscles favor reception of local muscle-derived dIlp6" (paragraph two, subheading “Muscle niche reactivates AMPs via dIlp6”). If technically possible, an improved 3-D view (reconstruction?) of AMP/Muscle contacts via filopodia would help non-expert readers to appreciate the positioning of AMPS relative to internal, median and external muscle* layers*. What happens to signaling if filipodia are disrupted? In conclusion, the authors do little to investigate the relevance of the filipodia in either homing or signaling. A detailed analysis is probably beyond the scope of this article but some attempt to clarify ideas about their function should be made, along the lines indicated above.*

Indeed a detailed analysis of homing appears beyond the scope of this article. However, to provide some insights into raised issues we performed series of experiments and provide now a more detailed view of AMPs localization, interaction with surrounding muscles and filopodia dynamics.

To better characterize AMPs localization with respect to internal, intermediate and external muscles (now schematized in revised Figure 1), we created Z-stack movies of stage 14 and stage 15 M6-gapGFP embryos (Video 2 and Video 3) stained for muscles and AMPs. We incorporated slice annotations to indicate position of AMPs with respect to muscle layers.

These movies clearly show that AMPs are not associated with one particular muscle layer. Also the ventral, lateral, dorso-lateral and dorsal AMPs have distinct internal/external locations. The lateral AMPs extend from the external to internal layer. The posterior lateral AMP (arrowhead in Z-stack movies) lies the most externally and is seen at the same optical level that the external lateral muscles (ExtLM: LT muscles). The anterior lateral AMP (arrow in Z-stack movies) lies more internally, mainly at the level of internal lateral muscles (IntLM: LO1 and SBM). The ventral AMPs are located in between external ventral muscles (ExtVM: VA1 and VA2) and intermediary ventral muscles (ImVM: VO3-VO6) but they send cellular extensions externally and are seen at the level of VA1 and VA2.

The dorso-lateral AMPs are clearly located under the external DT1 and lies mainly in between the intermediary dorsal muscles (InDM: DO3 and DO4) and internal dorsal muscles (IntDM: DA3). Finally, the dAMPs are located in between the external (ExtDM: DO1 and DO2) and internal dorsal muscles (IntDM: DA1, DA2).

Note: Please view movies (Video 2 and Video 3) frame by frame to appreciate AMPs positioning and to see corresponding annotations. Corresponding comments are included in the Results section.

We also made additional αPS1 integrin staining at three different developmental time points to better document integrin distribution associated with the filopodia and with the AMP cell bodies. This is presented as Z-stack projections of lateral AMPs at three different time points (new Figure 2’) and Z-stack 2 and 3 channels movies showing punctate decoration of lateral AMPs by αPS1 integrin at stage 16 (Figure 2–figure supplement 1A and 1B). We observed that the first αPS1 dotty signals associated with AMPs appear at late stage 14 (new Figure 2’) and are progressively enriched at stage 15 and 16 (new Figure 2’). We observe punctate αPS1 pattern associated mainly with AMP cell bodies but also aligned with filopodia (clearly seen at stage 16, arrows in new Figure 2’).

Similar but more discrete βPS pattern is also observed (new Figure 2’). The AMP cell bodies and filopodia-associated αPS1 dots are also shown in Z stack movies (Video 5 and Video 6) – view 2 and 3 channels movies in parallel frame by frame to follow αPS1 dots associated with the AMPs and corresponding annotations. As filopodia are highly dynamic structures we hypothesize that the filopodia-associated αPS1 dots mark cellular extensions that are getting in contact with target muscles and are in the process of stabilization. This possibility is supported by the observation that a subset of filopodia get indeed stabilized (arrowheads in Video 4).

Corresponding comments were included to the revised text, in the Results section.

To characterize filopodia dynamics we performed lifetime imaging of M6-gapGFP labeled AMPs at embryonic stage 15 (Video 4). The chosen developmental time window corresponds to the homing period in which AMPs actively send filopodia and try to contact target muscles. Z stacks were taken each 1 min during the period of 35 minutes. Leica SP8 confocal microscope with fast resonant scanner was used for live imaging. ImageJ was used to register and to annotate the movie.

We focused on and annotated the lateral AMPs. To follow the number of filopodia and the direction of their projection we labeled the extremities of all filopodia at each time point (indicated by yellow open circles, Video 4). This revealed that the lateral AMPs send filopodia non-randomly and mainly in anterior-dorsal directions, which correlate with location of SBM and LO/LT muscles to which lateral AMPs are connected by stage 16. We observe that between 6 and 9 filopodia are produced at each time point over the lifetime experiment. Interestingly, some of filopodia/cellular extensions are more stable than others (indicated by arrowheads). The anterior lateral AMP extension that appears stabilized and persists until the end of movie (arrowhead) ensures connection to LO1 and LT muscles. Filopodia projecting in dorsal direction (denoted by the arrowhead) potentially connecting the SBM persists few minutes but does not acquire stable state.

Note: A frame by frame viewing of the movie will allow to count number of filopodia per time point and appreciate filopodia extension and retraction events.

We included the corresponding comments in the Results section.

As suggested by the reviewers, we also attempted to affect filopodia and to test whether they play a role in signal transduction from muscles to the AMPs. To interfere with filopodia formation we tested several genetic contexts including overexpression of dominant negative form of CDC42 (CDC42DN) or Rac1 (Rac1DN), attenuation of Diaphanous (Diaphanous RNAi) and attenuation of DAAM (DAAM RNAi) (data not shown). Among them we found that AMP-targeted attenuation of DAAM encoding one of the formins produces the most penetrant phenotypes. In DAAM RNAi context we observed that AMP filopodia length is significantly reduced in 2^nd^ instar larvae (new Figure 7) and AMP cell number decreased when counted in mid-3^rd^ instar (new Figure 7) thus suggesting affected AMPs reactivation.

These data support a view that filopodia play an important role in AMPs reactivation.

2) A second point concerns the Gal4 drivers used. The authors’ statement that muscles are the source of Dilp6 acting on AMPs, is based on using a pan-mesodermal Mef2 driver (Figure 6). This driver could also affect the AMPS. An additional specific muscle driver e.g. MHC-Gal4 should be used. Furthermore, since drivers exist that are only active in subsets of somatic muscles, some contacted by AMPs, others not, use of these would be informative as to whether direct contacts between specific muscles and AMPs are instructive.

We previously documented (Jagla et al., Development 1998, 125, 3699-708) that *Mef2* gene expression is excluded from the Twist-expressing AMPs and used here Mef-Gal4 driver based on this observation. In response to the reviewers’ comment, we however tested Mef-Gal4 expression by crossing it to UAS-GFP. We observed that, like *Mef2* gene, the Mef-GAL4 driver is strongly expressed in somatic muscles of second instar larvae but is not active in AMPs (documentation of Mef-GAL4 expression not shown). Thus we are confident that data presented in Figure 7 are resulting from somatic muscle activity of Mef-Gal4.

Regarding drivers expressed in subsets of muscles such as Lb-Gal4 or Slou-Gal4 (we currently use in the lab for targeting individual muscles), they drive expression at relatively high levels in embryos but their activity drops down in larval stages and are thus not well adapted for studying muscle derived signals in larval stages. Moreover, the Slou-Gal4 drives also expression in ventral AMPs and Lb-Gal4 in lateral AMPs.

*vM6-Gal4 is generated from a Notch dependent enhancer. This complicates the experiments when Notch is manipulated. For example, inhibition of Notch would reduce expression of M6-Gal4. This can in turn impact on expression of the other RNAi's and modify the phenotypes. At least some experiments should be repeated with an independent Gal4 to verify the effects.*

M6-Gal4 driver has been generated using previously described (Lai et al., Development 2000, 127, 3441-55) 2.1 kb m6 regulatory element covering -2098 to +37 of m6 and carrying 4 putative Su(H) binding sites. This element has been previously shown to drive expression of GFP specifically in embryonic and larval AMPs (Lai et al., Development 2000, 127, 3441-55). We agree with the reviewers that M6-Gal4 is not a perfect tool for manipulating Notch, however we observed that it keeps active in the context Notch is attenuated indicating that Notch is not the only regulator of its activity and/or that a low level of Notch is sufficient to maintain M6-Gal4 active.

The following observations are in support of this view:

1) After crossing M6-Gal4 with UAS-NotchRNAi line we observed a strong decrease of AMP proliferation (Figure 4) indicating that Notch was attenuated and suggesting that M6 regulatory sequences are robust enough to keep active in the context of reduced Notch.

2) We tested activity of m6 regulatory element by imaging M6-gapGFP (GFP driven by the same 2.1 kb element) in wt AMPs (lacZ context) and in AMPs with reduced Notch (Notch RNAi context). Representative images showing anterior cluster of lateral AMPs acquired with the same confocal settings with a graph of fluorescence intensity are shown in Figure 4—figure supplement 2. This comparison reveals that attenuation of Notch does not lead to the loss of 2.1 kb m6 element activity. Measuring of fluorescence intensity reveals mean value reduction of about 10%.

Thus we are confident that the attenuation of Notch has no major effect on M6 driver activity.

Regarding alternative AMP drivers, to our knowledge, the only one reported so far is the 1151-Gal4 (Roy and VijayRaghavan, Development 1997, 124, 3333-41; Anant et al., Development 1998, 125, 1361-9). It is expressed in AMPs associated to 3^rd^ instar wing and leg imaginal discs and it has also been reported (Sudarsan et al., Dev Cell 2001, 1, 829-39) that 1151-Gal4 is expressed from 1^st^ instar in AMPs and in the fat body. Thus, 1151-Gal4 appears activated later than M6-Gal4 and its expression is not restricted to AMPs.

On the other hand, 1151-Gal4 is an enhancer trap insertion, mapped to cut locus encoding a homeodomain transcription factor regulated by Notch (Sun and Deng, Development 2005, 132, 4299-308). It is thus difficult to exclude Notch influence on 1151-Gal4, which seems not to be a good alternative for driving expression specifically in AMPs and in a Notch independent way.*3) The authors argue that the effects of Notch involve ligand-independent signaling, but they have not really tested that. Dx and Su(Dx) can also affect ligand dependent signaling. A strategy that blocks ligand dependent signaling would clarify this issue e.g. by blocking the ADAM protease.*

We agree with the reviewers that we did not test all players in ligand-independent Notch activation and that Deltex can also affect ligand dependent signaling. In consequence we modified our statements in the text from “ligand independent Notch signaling” to “Deltex-involving Notch signaling”.

Regarding Deltex, we revise our statement in the text (“To further explore the link between Insulin and Notch pathway we tested expression of the ubiquitin ligase Deltex, known to be involved in ligand-independent intracellular activation of Notch by promoting its mono-ubiquitinated state (Hori et al., J Cell Biol 2011, 195, 1005-15)”).

Regarding, blocking ligand dependent Notch signaling, we already did it using a dominant negative allele of Notch called ECN and consisting of extracellular and transmembrane domains but lacking all the intracellular Notch sequence (Rebay et al., Cell 1993, 74, 319-29; Klein et al., Dev Biol 1997, 189, 123-34; please see revised version Figure 6—figure supplement 1). Overexpression of ECN was previously found to mimic loss of function phenotypes of Notch (Rebay et al., Cell 1993, 74, 319-29) and it was then used in more than 50 publications for mimicking/testing loss of canonical Notch signaling. We found that overexpression of ECN in AMPs has no effect on their reactivation supporting ligand-independent Notch activation. A notion that is also consistent with the increased number of AMPs after attenuation of Shrub (Figure 6).

Regarding blocking Kuzbanian/Adam10 protease we followed the reviewers’ suggestion and overexpressed a dominant negative form of Kuzbanian (stock BL6578) in AMPs. We found that blocking Kuzbanian function has a minor influence on AMPs number (about 15% less of lateral AMPs counted in 30 segments compared to control). This supports the view that Kuzbanian-involving ligand dependent Notch signaling has no major role in AMPs reactivation.

However, whether Kuzbanian acts exclusively in ligand-dependent way remains unclear. To our knowledge the ligand independent Notch activation also involves series of cleavages and in particular the gamma-secretase-involving cleavage of Notch-derived NEXT-like substrate, normally generated upon S2 cleavage by Kuzbanian (Schneider et al., J Cell Sci 2013, 126, 645-56). Thus, with no clear view on Kuzbanian function in ligand-independent versus-ligand dependent Notch signaling we thought including AMP count data from M6>KuzDN context (data not shown) would not reinforce the already presented data.

The conclusion that N acts downstream of InR in stimulating AMP proliferation is novel and of potential strong impact in the field. Yet, the authors should leave open the possibility that N acts also independent of InR since overexpressing NICD induces higher number of AMPs (Figure 3 and Figure 4) than InRCA, although not higher Myc levels (Figure 4).

We agree with this comment and added the following statement: “However, because overexpressing of NICD induces higher number of AMPs than InRCA (Figure 6 and Figure 6—figure supplement 1) we cannot exclude a possibility that Notch also acts in an InR-independent way.”

[Editors' note: further revisions were requested prior to acceptance, as described below.] *1) On the one hand, the authors use the expression from the m6 enhancer (m6-GFP) expression to argue that Notch is active in the AMPs, but then they now argue that the expression from the same enhancer (in m6-Gal4) can't be reduced when Notch activity is attenuated. These claims are rather contradictory. Most likely there is a simple explanation (perdurance of Gal4 and GFP in the latter?) but this does make the results confusing for the reader (e.g. subheading “Notch acts downstream of the Insulin pathway and regulates the proliferation of AMPs via dMyc” and in the Discussion: "Here we followed the GFP expression driven by a Notch-responsive element of the m6 gene and found that the Notch pathway was active in dormant AMPs"). Can they be confident that this is reading out Notch activity at that time? Maybe the enhancer requires Notch for activation but not for maintenance? Given the data they have provided, I suggest that they tone down their statements.*

We agree that because of GaL4/GFP perdurance it is difficult to argue for activation versus maintenance of Notch in quiescent AMPs. As suggested, we toned down our statements in the Results and in the Discussion.

In the Results, the original sentence “As the decrease in AMP numbers in a Notch-RNAi context has not been associated with a reduced level of GFP driven by the same M6 regulatory element (Figure 4—figure supplement 2) we assume that the M6-Gal4 driver stays active in Notch attenuated conditions” now reads: “As the decrease in AMP numbers in a Notch-RNAi context has not been associated with a reduced level of GFP driven by the same m6 regulatory element (Figure 4—figure supplement 2), we hypothesize that a low Notch level is sufficient to maintain m6 activity but we cannot rule out a possibility that perdurance of Gal4 and GFP in larval stages plays role as well.”

In the Discussion, the passage “Here we followed the GFP expression driven by a Notch-responsive element of the *m6* gene and found that the Notch pathway was active in dormant AMPs…” now reads: “Here we followed the GFP expression driven by a Notch-responsive element of the *m6* gene and found that dormant AMPs are GFP-positive, which suggests that like in satellite cells, Notch is involved in setting the AMP quiescent state.”

*2) Teasing apart the ligand dependent versus ligand independent Notch activity is tricky. The authors discuss in the rebuttal their experiment with the KuzDN, which gives minor effects that they argue should be ignored. But at the same time they place quite a strong emphasis on the role of Deltex despite the fact that the results in this section are confusing and contradictory (e.g. over-expressing and attenuating Deltex have the same effects, likewise Su(dx), likewise kurtz). Given these contrary data, it doesn't seem appropriate to make major conclusions about the roles of these factors so I suggest that they remove the scheme from the right hand part of Figure 8 and further moderate their statements about the role of Dx. Altering Deltex activity does influence the AMPS but it is not clear that it is the main effector of the signaling through InR nor the critical regulator of Notch in this context.*

As pointed by the reviewers, the fact that over-expressing and attenuating Deltex, Su(Dx) and Kurtz leads to the similar AMP overproliferation phenotype is surprising even if our interpretation based on stoichiometry of Kurtz and Deltex could explain this paradigm. In consequence, we removed the corresponding scheme from Figure 8 and moderated our statement concerning Deltex and its role as InR effector.

We modified the following sentence in the Results section: “Taken together, this body of evidence points to the conclusion that during AMP reactivation, the Insulin pathway activates Notch in a Deltex and Shrub-involving ligand-independent way”. This now reads: “Taken together, this body of evidence suggests that during AMP reactivation, the Insulin pathway activates Notch in a Deltex and Shrub-involving ligand-independent way.” *3) Figure 6—figure supplement 1 presents a hypothetical scheme not directly connected to experiments described in the manuscript. It should be removed.*

We agree that the scheme is not related to experimental data but rather to hypotheses that were tested. As suggested, we removed it from Figure 6—figure supplement 1.